# Sequence structure organizes items in varied latent states of working memory neural network

**Qiaoli Huang[1,2,3], Huihui Zhang[1,2,3], Huan Luo[1,2,3]***

[1]School of Psychological and Cognitive Sciences, Peking University, Beijing, China; [2]PKU-IDG/McGovern Institute for Brain Research, Peking University, Beijing, China; [3]Beijing Key Laboratory of Behavior and Mental Health, Peking University, Beijing, China

**Abstract** In memory experiences, events do not exist independently but are linked with each other via structure-based organization. Structure context largely influences memory behavior, but how it is implemented in the brain remains unknown. Here, we combined magnetoencephalogram (MEG) recordings, computational modeling, and impulse-response approaches to probe the latent states when subjects held a list of items in working memory (WM). We demonstrate that sequence context reorganizes WM items into distinct latent states, that is, being reactivated at different latencies during WM retention, and the reactivation profiles further correlate with recency behavior. In contrast, memorizing the same list of items without sequence task requirements weakens the recency effect and elicits comparable neural reactivations. Computational modeling further reveals a dominant function of sequence context, instead of passive memory decaying, in characterizing recency effect. Taken together, sequence structure context shapes the way WM items are stored in the human brain and essentially influences memory behavior.

***For correspondence:**
huan.luo@pku.edu.cn

## Introduction

Working memory (WM), more than just passively maintaining inputs, is also an active process that modifies and even reorganizes information representations to guide future behavior (*Baddeley, 2003*). For instance, retro-cues during retention could enhance WM performance (*Griffin and Nobre, 2003*; *Landman et al., 2003*; *Larocque et al., 2014*) and modulate neural responses in different brain regions (*Christophel et al., 2018*; *Yu et al., 2020*), suggesting that attention could flexibly modulate information that has already been maintained in WM (*Myers et al., 2017*). In addition to top-down attention, contexts and structure in which the to-be-memorized items are embedded also influence memory performance (*Brady et al., 2011*; *Brady and Tenenbaum, 2013*; *DuBrow and Davachi, 2013*; *Gershman et al., 2013*; *Jiang et al., 2000*; *Oberauer and Lin, 2017*). A typical example is the serial position effect for sequence memory, that is the recently presented item shows better memory performance compared to early one (*Broadbent and Broadbent, 1981*; *Burgess and Hitch, 1999*; *Gorgoraptis et al., 2011*; *Huang et al., 2018*; *Jones and Oberauer, 2013*). However, it remains largely unknown how structure context modulates or reorganizes the way multiple WM items are represented and maintained in the human brain. Here, we particularly focused on sequence structure, an essential one that mediates many cognitive functions, for example sequence memory, speech, movement control (*Davachi and DuBrow, 2015*; *Giraud and Poeppel, 2012*; *Polyn et al., 2009*).

Previous neural recordings demonstrate that a sequence of items would elicit serial and temporally compressed reactivations, which might reflect a memory consolidation process that reorganizes the incoming inputs (*Bahramisharif et al., 2018*; *Foster and Wilson, 2006*; *Huang et al., 2018*;

*Kurth-Nelson et al., 2016*; *Liu et al., 2019*; *Siegel et al., 2009*; *Skaggs and McNaughton, 1996*). Interestingly, recent modeling and empirical studies propose that, in addition to maintenance via sustained or serial reactivations (*Curtis and D'Esposito, 2003*; *Goldman-Rakic, 1995*; *Vogel and Machizawa, 2004*), information could also be stored in synaptic weights of the network, that is activity-silent view (*Miller et al., 2018*; *Mongillo et al., 2008*; *Rose et al., 2016*; *Sprague et al., 2016*; *Stokes, 2015*; *Trübutschek et al., 2017*; *Wolff et al., 2017*). In other words, multiple items could be maintained in latent or hidden states of the WM neural network.

How to access the WM information stored in the 'activity-silent' network? An impulse-response approach, by presenting a PING stimulus to transiently perturb the WM system, efficiently reactivates WM representations (*Wolff et al., 2017*). Moreover, attended but not unattended item is successfully reactivated, implying that top-down attention modulates the latent states WM items reside in *Wolff et al., 2017*; *Wolff et al., 2020*. Here, we used this method to assess whether a list of WM items of equal task relevance would be reorganized by imposed sequence structure in varied latent states and in turn show distinct reactivation profiles. The hypothesis is also motivated by our previous findings demonstrating backward reactivations for sequence memory, which implies a more excitable state for recent vs. early items (*Huang et al., 2018*).

We recorded magnetoencephalography (MEG) signals when human subjects (N = 24) retained a sequence of orientations and their ordinal positions. A time-resolved multivariate inverted encoding model (IEM) (*Brouwer and Heeger, 2009*; *Brouwer and Heeger, 2011*; *Sprague et al., 2014*) was used to reconstruct the neural representations of each orientation (i.e. 1st item, 2nd item) over time. Importantly, a nonspecific PING stimulus (i.e. impulse) was presented during retention, aiming to transiently perturb the WM network so that the latent states of the stored items could be accessed. Our results demonstrate a backward reactivation profile such that the impulse triggers the neural representation of the 2nd item first, followed by that of the 1st item, thus supporting their different latent states. Moreover, the neural reactivation pattern well predicts the recency effect in behavior. In contrast, in another MEG experiment when subjects (N = 24, new subjects) retained the same sequence without needing to memorize the ordinal structure, the two orientations show similar reactivation profiles. Finally, computational modeling demonstrates that the sequence contexts, instead of passive memory decay, largely characterizes the recency behavior. Our findings constitute converging evidence supporting a central function of sequence structure in WM via reorganizing multiple items in the brain (i.e. in varied latent states) and influencing memory behavior. Generally speaking, our findings provide new perspectives for the neural mechanism underlying task-related multi-item information storage in the WM system.

## Results

### Experimental procedure and recency behavior (Experiment 1)

Twenty-four subjects participated in Experiment 1 and their brain activities were recorded using a 306-channel magnetoencephalography (MEG) system (Elekta Neuromag system, Helsinki, Finland). As shown in *Figure 1A*, each trial consists of three periods – encoding, maintaining, and recalling. During the encoding period, participants viewed two serially presented grating stimuli and were instructed to memorize the two orientations as well as their order (1st and 2nd orientations). After a 2 s maintaining period, a retrospective cue (retro-cue) appeared to instruct subjects which item (1st or 2nd) would be tested. Next, a probe grating was presented and participants indicated whether the orientation of the probe was rotated clockwise or anticlockwise relative to that of the cued grating. Note that since the retro-cue appeared only during the recalling period, subjects would need to hold the two WM orientations simultaneously in WM throughout the retention interval. Critically, during the maintaining period, a high-luminance PING stimulus that does not contain any orientation information was presented, aiming to transiently perturb the WM network so that the stored information and its associated latent states could be assessed.

As shown in *Figure 1B*, the memory behavioral performance exhibited the typical recency effect, that is $2^{nd} > 1st$ item (N = 24, 1st item: mean = 0.77, s.e. = 0.012; $2^{nd}$ item: mean = 0.79, s.e = 0.013; paired t-test, df = 23, t = 2.18, p = 0.039, Cohen's d = 0.45), consistent with previous findings (e.g. *Broadbent and Broadbent, 1981*; *Burgess and Hitch, 1999*; *Gorgoraptis et al., 2011*; *Huang et al., 2018*; *Jones and Oberauer, 2013*). When plotting the psychometric function of the

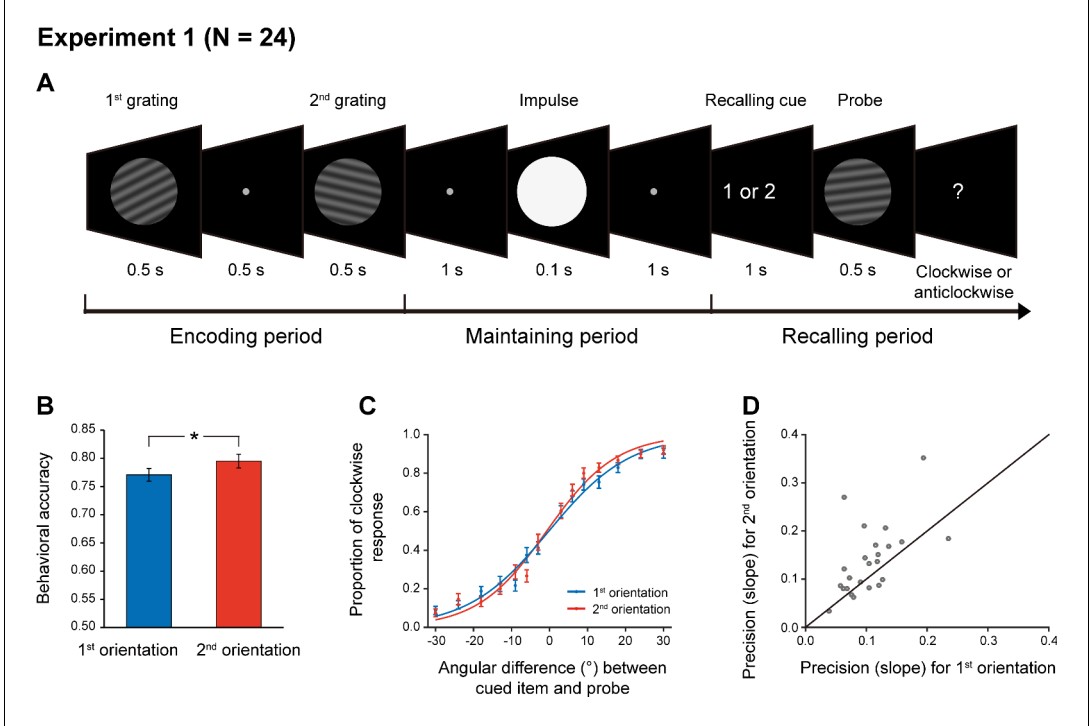

**Figure 1.** Experimental paradigm and recency effect (Experiment 1). (A) Experiment 1 paradigm (N = 24). Subjects were first sequentially presented with two grating stimuli (1st and 2nd gratings) and needed to memorize the orientations of the two gratings as well as their temporal order (1st or 2nd orientation). During the maintaining period, a high luminance disc that does not contain any orientation information was presented as a PING stimulus to disturb the WM neural network. During the recalling period, a retro-cue first appeared to instruct subjects which item (1st or 2nd) would be tested. Next, a probe grating was presented and participants indicated whether the orientation of the probe was rotated clockwise or anticlockwise relative to that of the cued grating. (B) Grand average (mean ± SEM) behavioral accuracy for the 1st (blue) and 2nd (red) orientations. (C) Grand average (mean ± SEM) psychometric functions for the 1st (blue) and 2nd (red) items as a function of the angular difference between the probe and cued orientation. Note the steeper slope for the 2nd vs. 1st orientation, that is recency effect. (D) Scatter plot for the slope of the psychometric function for the 1st (x axis) and 2nd orientations (y axis). (*: p < 0.05).

The online version of this article includes the following source data for figure 1:

**Source data 1.** Source data for *Figure 1*.

proportion of clockwise response as a function of angular difference between the probe and the cued WM grating, the 2nd item showed steeper slope than the 1st item (*Figure 1C*; 1st item: mean = 0.11, s.e. = 0.009; 2nd item: mean = 0.14, s.e = 0.015; paired t-test, df = 23, t = 2.64, p = 0.015, Cohen's d = 0.54). This pattern could be reliably observed for individual subjects (*Figure 1D*), consistently supporting the recency effect.

## Time-resolved neural representations of orientation features (Experiment 1)

We used a time-resolved inverted encoding model (IEM) (*Brouwer and Heeger, 2009*; *Brouwer and Heeger, 2011*; *Sprague et al., 2014*) to reconstruct the neural representations of orientation features at each time point throughout the experimental trial. We first verified this method by applying it to the encoding period when the to-be-memorized grating stimuli were presented physically. Specifically, the orientation decoding performance was characterized by reconstructed channel response as a function of angular difference between an orientation-of-interest and other orientations (see details in Materials and methods). If MEG signals do carry information about specific orientation, the reconstructed channel response would reveal a peak at center and gradually decrease on both sides. *Figure 2AB* plot reconstructed channel responses for the 1st and 2nd WM orientations, respectively, as a function of time throughout the encoding phase. It is clear that right after the presentation of the 1st grating, the reconstructed channel response for the 1st orientation

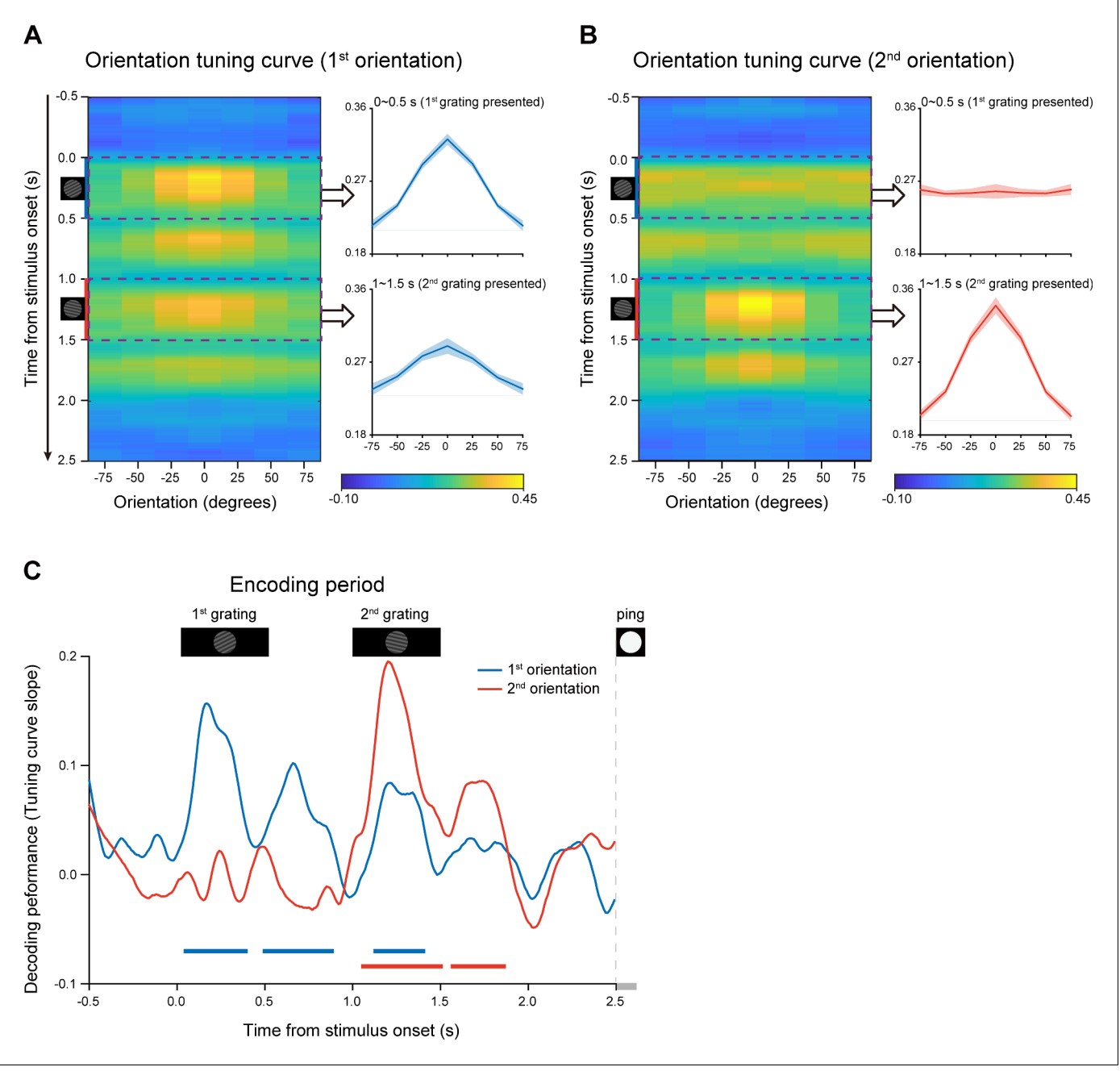

**Figure 2.** Time-resolved orientation representations during encoding period (Experiment 1). An IEM was used to reconstruct the neural representation of orientation features, characterized as a population reconstructed channel response as a function of channel offset (x-axis) at each time bin (y-axis). Successful encoding of orientation features would show a peak at center around 0 and gradually decrease on both sides, whereas lack of orientation information would have a flat channel response. The slope of the channel response was further calculated (see Materials and methods for details) to index information representation. (A) Left: Grand average time-resolved channel response for the 1st orientation (orientation of the 1st WM grating) throughout the encoding period during which the 1st and 2nd gratings (small inset figures on the left) were presented sequentially. Right: grand average (mean ± SEM) channel response for the 1st orientation averaged over the 1st grating presentation period (0–0.5 s, upper) and the 2nd grating presentation (1–1.5 s, lower). (B) Left: Grand average time-resolved channel response for the 2nd orientation (orientation of the 2nd WM grating) during the encoding period. Right: grand average (mean ± SEM) channel response for the 2nd orientation averaged over the 1 st (0–0.5 s, upper) and 2nd (1–1.5 s, lower) grating presentation period. (C) Grand average time courses of the channel response slopes for the 1st (blue) and 2nd (red) orientations during the encoding period. Horizontal lines below indicate significant time ranges (cluster-based permutation test, cluster-defining threshold p < 0.05, corrected significance level p < 0.05) for the 1st (blue) and 2nd (red) orientations.

The online version of this article includes the following source data and figure supplement(s) for figure 2:

*Figure 2 continued on next page*

*Figure 2 continued*

**Source data 1.** Source data for *Figure 2*.
**Figure supplement 1.** Time-resolved reconstructed channel responses.

showed central peak (*Figure 2A*), whereas that for the 2nd orientation displayed information representation only after the presentation of the 2nd grating (*Figure 2B*). To further quantify the time-resolved decoding performance, the slope of the reconstructed channel response was estimated at each time point in each trial, for the 1st and 2nd orientations, respectively (see details in Materials and methods). As shown in *Figure 2C*, the 1st orientation (blue) showed information

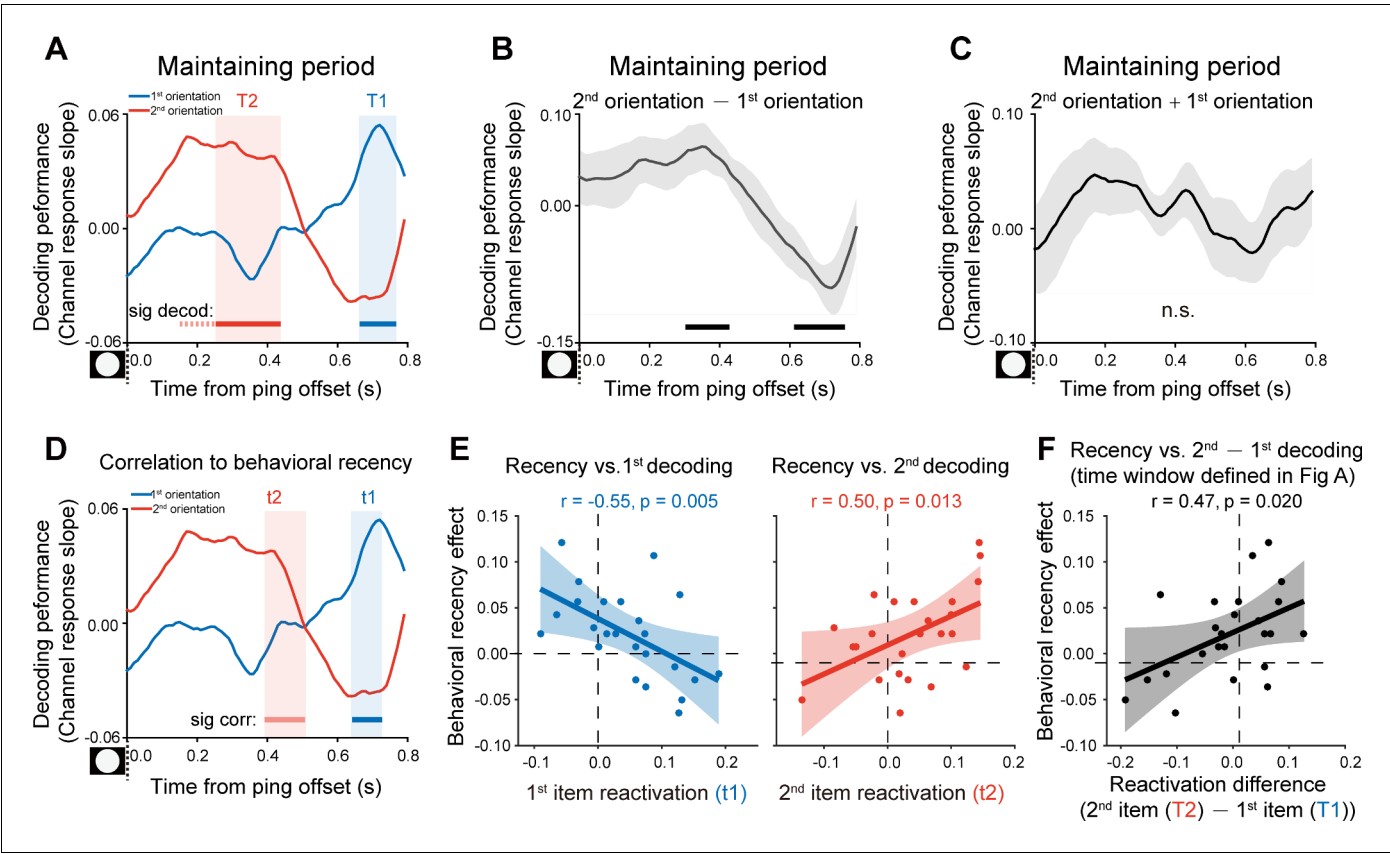

**Figure 3.** Backward reactivations during retention and behavioral relevance (Experiment 1). (**A**) Grand average time courses of the channel response slope for the 1st (blue) and 2nd (red) WM orientations after PING (inset in the bottom left) during the delay period. Horizontal lines below indicate time ranges of significant decoding strengths for the 1st (blue, T1) and 2nd (red, T2) orientations, respectively. (**B**) Grand average (mean ± SEM) time course of the difference between the 1st and 2nd channel response slopes (2nd – 1st) and the significant time points. (**C**) Grand average (mean ± SEM) time course of the sum of the 1st and 2nd channel response slopes (1st + 2nd). (**D**) Subject-by-subject correlations between the decoding performance and the recency effect were calculated at each time point. Horizontal lines below indicate time points of significant behavioral correlations (Pearson's correlation after multi-comparison correction) for the 1st (blue) and 2nd (red) items, respectively. (**E**) Left: scatterplot (N = 24) of recency effect vs. 1st decoding strength averaged over t1 (0.67–0.72 s after PING). Right: scatterplot of recency effect vs. 2nd decoding strength averaged over t2 (0.4–0.43 s after PING). (**F**) Scatterplot (N = 24) of recency effect vs. decoding difference (2nd at T2 – 1st at T1). Each dot represents an individual subject. (Horizontal solid line: cluster-based permutation test, cluster-defining threshold p < 0.05, corrected significance level p < 0.05; Horizontal light solid line: marginal significant, cluster-defining threshold p < 0.05, 0.05 < cluster p < 0.1); Horizontal dashed line: marginal significance, cluster-defining threshold p < 0.1, 0.05 < cluster p < 0.1. Shadow indicates 95% confidence interval.

The online version of this article includes the following source data and figure supplement(s) for figure 3:

**Source data 1.** Source data for *Figure 3*.
**Figure supplement 1.** Neural-recency correlation and confidence bands for decoding time courses.
**Figure supplement 2.** Cross-generalization results (Exp.1).
**Figure supplement 3.** Decoding performance based on alpha-band power.

representation right after the 1st grating (0.05–0.39 s, corrected cluster p = 0.002; 0.5–0.88 s, corrected cluster p < 0.001; 1.13–1.4 s, corrected cluster p = 0.002), and neural representation of the 2nd orientation (red) emerged after the 2nd grating (1.06–1.5 s, corrected cluster p < 0.001; 1.57–1.86 s, corrected cluster p = 0.008). Therefore, orientation information could be successfully decoded from MEG signals in a time-resolved manner. Moreover, the decoding performance for both the 1st and 2nd orientations gradually decayed to baseline, around 0.5 s after the offset of the 2nd item, suggesting that the WM network now entered the activity-silent WM states (*Rose et al., 2016*; *Stokes, 2015*; *Wolff et al., 2017*). It is notable that nonsignificant decoding results do not exclude sustained firing at neuronal level (*Miller et al., 2018*), given the limited sensitivity of MEG and EEG signals.

## PING stimulus elicits backward reactivations during retention (Experiment 1)

After confirming the decoding approach during the encoding period, we next used the same analysis to examine the orientation representations held in WM that would be presumably reactivated by the PING stimulus during retention (see *Figure 1A*). *Figure 3A* plots the decoding performances for the 1st (blue) and 2nd (red) WM orientation features, as a function of time after the PING stimulus (see the reconstructed channel response in *Figure 2—figure supplement 1AB*). Interestingly, instead of being activated simultaneously, the 1st and 2nd orientations showed distinct temporal profiles, that is the 2nd orientation showed earlier activation (T2: from 0.26 to 0.43 s, corrected cluster p = 0.011) than the 1st orientation (T1: from 0.67 to 0.76 s, corrected cluster p = 0.030), with approximately 0.3 s temporal lag in-between. To further verify their distinct patterns, we computed the difference (*Figure 3B*) and sum (*Figure 3C*) between the 1st and 2nd decoding temporal profiles. The difference course was significant (0.31–0.42 s, corrected cluster p = 0.023; 0.62–0.75 s, corrected cluster p = 0.013) (*Figure 3B*), while their sum did not show any significance (*Figure 3C*), further confirming that the two items were activated at different latencies. The results suggest that the 1st and 2nd items, instead of residing in equally excitable states, tend to be stored in different latent states of the WM network. As a consequence, a transient perturbation of the network would produce an early 2nd item reactivation and a late 1st item response.

## Reactivation profiles correlate with recency effect (Experiment 1)

We next evaluated the behavioral relevance of the reactivation profiles on a subject-by-subject basis, by relating the decoding strength to the recency effect, at each time point. As shown in *Figure 3D*, both the 1st (blue) and 2nd (red) item reactivations correlated with the recency effect (horizontal lines, blue for the 1st item and red for the 2nd item), but at different time (marginally significant, 1st item: 0.65–0.72 s, corrected cluster p = 0.055, blue shades; 2nd item: 0.4–0.5 s, corrected cluster p = 0.038, red shades) (see the raw correlation coefficient time course in *Figure 3—figure supplement 1A*). Note that the temporal windows (t1, t2 in *Figure 3D*) showing significant neural-behavioral correlations was defined independent of the reactivation strength (*Figure 3A*). *Figure 3E* illustrates the correlation scatterplots within the two time windows (i.e. t1, t2) that were defined in *Figure 3D*, respectively. Specifically, the recency effect covaried positively with the 2nd item (*Figure 3E*, right panel; Pearson's correlation, N = 24, r = 0.50, p = 0.013) and negatively with the 1st item (*Figure 3E*, left panel; Pearson's correlation, N = 24, r = −0.55, p = 0.005). Moreover, we chose time bins solely based on significant reactivations regardless of its relevance to recency effect (*Figure 3A*; T1 for the 1st item, blue shades; T2 for the 2nd item, red shades). As shown in *Figure 3F*, consistently, the 2nd – 1st reactivation difference was correlated with the recency effect (Pearson's correlation, N = 24, r = 0.47, p = 0.02). Taken together, the backward reactivation profiles that index distinct latent states in WM, show strong relevance to memory behavior, that is stronger recency effect is accompanied by larger, early 2nd item reactivation and weaker, late 1st item reactivation during the delay period.

## Experimental procedure and weakened recency effect (Experiment 2)

One possible reason for the backward reactivation in Experiment 1 is that the 2nd item enters the memory system later than the 1st item and thus decays less, leading to a more excitable state for the lately presented item. In other words, the distinct latent states might solely arise from their

different passive memory traces left in the network. To address the issue, we performed Experiment 2 using the same stimuli and paradigm as Experiment 1, except that subjects did not need to memorize the temporal order of the two orientations. Specifically, as shown in *Figure 4A*, subjects viewed two serially presented gratings and were instructed to memorize the two orientations. During the recalling period, instead of indicating the 1st or 2nd orientation, a retro-cue appeared to instruct subjects which item that has either smaller or larger angular values (relative to a vertical axis in a clockwise direction) would be tested later. Next, a probe grating was presented and participants indicated whether the orientation of the probe was rotated clockwise or anticlockwise relative to the cued grating. Thus, if the different latent states are due to the passive decay of the serially presented items, we would expect similar recency effect as well as backward reactivation as shown in Experiment 1.

Twenty-four new subjects participated in Experiment 2. Interestingly, the 1st and 2nd items showed similar memory performance (*Figure 4B*; 1st item: mean = 0.77, s.e. = 0.011; 2nd item: mean = 0.78, s.e = 0.011; paired t-test, df = 23, t = 1.57, p = 0.13, Cohen's d = 0.32), and the psychometric functions did not differ in slopes for the 2nd and 1st orientations (*Figure 4CD*; N = 24, 1st item: mean = 0.11, s.e. = 0.008; 2nd item: mean = 0.12, s.e = 0.009; paired t-test, df = 23, t = 1.38, p = 0.18, Cohen's d = 0.28). Thus, recency effect tends to be weakened when the ordinal structure is not needed to be retained in WM.

Moreover, to confirm that subjects memorized two independent orientations as instructed rather than their relative angle, we fitted a generalized linear mixed-effects model to behavior. The results showed that only the angular difference between the to-be-retrieved orientation and the probe accounted for the behavioral performance ($\beta$ = 0.0013, t = 3.45, p < 0.001), whereas the angular difference between the to-be-memorizedorientations did not ($\beta$ < 0.0001, t = 0.50, p = 0.62).

## Disrupted backward reactivations during retention (Experiment 2)

We used the same IEM approach to reconstruct the time-resolved neural representations of orientation features at each time point in Experiment 2. First, the encoding period showed a similar pattern as Experiment 1 (*Figure 4E*; also see the reconstructed channel response in *Figure 2—figure supplement 1CD*). Specifically, the decoding performance of the 1st (blue) and 2nd (red) orientations displayed successive temporal profiles that were time-locked to the presentation of the corresponding grating stimuli (1st item: 0.07–0.32 s, corrected cluster p = 0.005; 0.41–0.85 s, corrected cluster p = 0.002; 1.09–1.32 s, corrected cluster p = 0.005; 2nd item: 1.11–1.86 s, corrected cluster p < 0.001). Combining Experiment 1 and Experiment 2 revealed a clear serial profile during the encoding period (*Figure 4F*; 1st item: 0.04–0.89 s, corrected cluster p < 0.001; 1.09–1.41 s, corrected cluster p = 0.003; 2nd[d] item: 1.06–1.88 s, corrected cluster p < 0.001).

In contrast, the reactivation profiles (marginally significant trend) during the delay period in Experiment 2 (*Figure 4G*; 0.1–0.2 s; 1st item, p = 0.056, one-tailed; 2nd item, p = 0.019, one-tailed) did not show the backward pattern as observed in Experiment 1 (see the reconstructed channel response in *Figure 2—figure supplement 1EF*). Consistently, the difference course between the 1st and 2nd decoding performances did not reach statistical significance (*Figure 4H*); corrected cluster p > 0.5. Interestingly, different from Experiment 1 (*Figure 3C*), the sum of the 1st and 2nd decoding performance showed significant responses (*Figure 4I*; 0.1–0.2 s, corrected cluster p = 0.024), somewhat supporting their common reactivation profiles. Finally, as shown in *Figure 4J*, the 2nd – 1st reactivation difference (i.e., backward reactivation index) showed marginally significant difference between Experiment 1 and 2, within two temporal windows (0.35–0.42 s, independent t-test, cluster p = 0.058; 0.59–0.67 s, independent t-test, cluster p = 0.043). Furthermore, Experiment 2 did not show the reactivation-recency correlations either (*Figure 4K*), compared to Experiment 1 (*Figure 3—figure supplement 1B*).

Since Experiment 2 instructed subjects to maintain the two orientations in terms of 'big' or 'small' labels, WM representations might be organized based on different principles (i.e. big or small) rather than ordinal position in Experiment 1. Interestingly, decoding analysis based on the big/small labels in Experiment 2o again revealed similar reactivation profiles for the two orientations (*Figure 4—figure supplement 1*), suggesting that the new labeling could not reorganize items in varied latent states as sequence structure context does in Experiment 1. Finally, the neural representation of WM items could neither be generalized from the encoding to maintaining periods, nor across items in

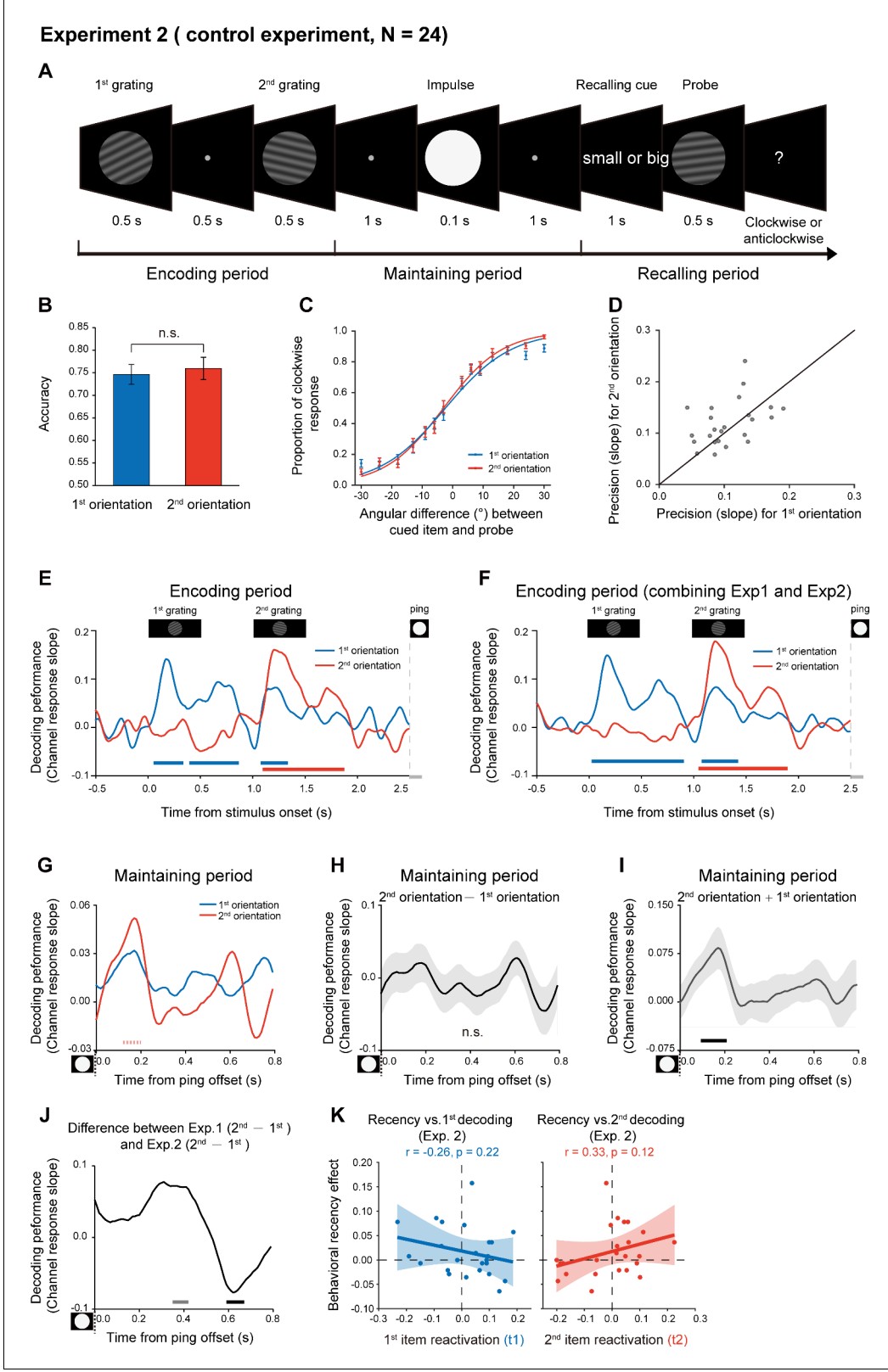

**Figure 4.** Experimental paradigm and results (Experiment 2). (**A**) Experiment 2 (N = 24) had the same stimuli and paradigm as Experiment 1, except that subjects did not need to retain the temporal order of the two orientation features. Subjects were first sequentially presented with two grating stimuli (1st and 2nd gratings) and needed to memorize the two orientations. During the recalling period, a retro-cue (small or big in character) appeared to instruct subjects which item that has either smaller or larger angular values relative to the vertical axis in a clockwise direction would be

*Figure 4 continued on next page*

*Figure 4 continued*

tested later. Next, a probe grating was presented and participants indicated whether the orientation of the probe was rotated clockwise or anticlockwise relative to that of the cued grating. (**B**) Grand average (mean ± SEM) behavioral accuracy for the 1st (blue) and 2nd (red) orientations. (**C**) Grand average (mean ± SEM) psychometric functions for the 1st (blue) and 2nd (red) orientations as a function of the angular difference between the probe and cued orientation. (**D**) Scatter plot for the slope of the psychometric function for the 1st (x axis) and 2nd (y axis) orientations. (**E**) Grand average time courses of the channel response slopes for the 1st (blue) and 2$^{nnd}$(red) orientations during the encoding period. Horizontal lines below indicate significant time points for the 1st (blue) and 2nd (red) orientations. (**F**) The same as E, but pooling Experiment 1 and Experiment 2 together during the encoding period (N = 48). (**G**) Grand average time courses of the channel response slope for the 1st (blue) and 2nd (red) WM orientations after PING (inset in the bottom left) during retention. (**H**) Grand average (mean ± SEM) time course of the difference between 1st and 2nd channel response slopes (2nd − 1st). (**I**) Grand average (mean ± SEM) time course of the sum of the 1st and 2nd channel response slopes (1st + 2nd) and significant time points. (**J**) Grand average time course of the 2nd − 1st reactivation difference between Exp 1 and Exp 2. (**K**) Same as *Figure 3E* but for Exp. 2. (Horizontal solid line: cluster-based permutation test, cluster-defining threshold p < 0.05, corrected significance level p < 0.05; Horizontal light solid line: marginal significant, cluster-defining threshold p < 0.05, 0.05 < cluster p < 0.1); Horizontal dashed line: marginal significance, cluster-defining threshold p < 0.1, 0.05 < cluster p < 0.1.

The online version of this article includes the following source data and figure supplement(s) for figure 4:

**Source data 1.** Source data for *Figure 4*.

**Figure supplement 1.** Time-resolved orientation decoding performances based on big/small label for Exp.1 and Exp.2.

the reactivations triggered by PING stimulus (*Figure 3—figure supplement 2*), further advocating that WM items embedded in the sequence structure context are reorganized in varied latent states.

Taken together, when the two serially presented items are maintained in WM without a sequence structure imposed on them, they tend to be stored in comparable latent states of WM network, thereby having similar probability to be activated after a transient impulse and showing relatively similar reactivation profiles and no associations to the recency behavior. The results thus weaken the alternative interpretation that it is the different passive memory decay that gives rise to the different latent states as observed in Experiment 1.

## Experiment 3 and computational model

Given the inherent time lag between the two sequentially presented items, the passive memory decay is seemingly a very straightforward interpretation for the recency effect. To further characterize the recency effect in terms of passive memory decay and sequence structure, we performed a behavioral experiment on 24 new subjects. Specifically, similar experiment design and task (retaining two orientations as well as their temporal order) as Experiment 1 (*Figure 1A*) were employed, except that there were three time intervals between the 2nd grating (1 s, 2.5 s, 4 s) and PING stimulus. Notably, Experiment 1 with fixed interval between the 2nd item and PING would make it difficult to reliably estimate the passive memory decay rate in behavioral performance, while the current design with three time lags would allow us to examine the passive memory decay and sequence context modulation in parallel from the same data set. As shown in *Figure 5A*, significant main effects in both serial position (i.e. recency effect; $F_{(1,23)} = 7.03$, $p = 0.014$, $\eta^2 = 0.23$) and memory decay ($F_{(2,23)} = 5.21$, $p = 0.009$, $\eta^2 = 0.19$) were observed in the behavioral experiment, and there was no interaction effect ($F_{(2,23)} = 2.04$, $p = 0.14$, $\eta^2 = 0.08$). The results thus confirm that memory performance of items in a list would be determined by both passive memory decay and their positions in the sequence (e.g. 1st or 2nd).

We next built a computational model that comprises sequence structure ($\sigma_1$ and $\sigma_2$ for the 1st and 2nd items, separately) and passive memory decay ($\beta$) to assess their respective contribution to the recency effect, considering that the standard deviation of orientation representation in working memory increases linearly with delay duration (*Shin et al., 2017*). Here, for an item at a given time t after being encoded into WM, the standard deviation of its representational noise was set to be $\sigma + \beta t$. The parameter $\beta$ represents the memory decay rate and the parameter $\sigma$ refers to the initial standard deviation of orientation representational noise at t = 0, whose value is either $\sigma_1$ (1st item) or $\sigma_2$ (2nd item). Since the 1st item appears prior to the 2nd item, it would have a longer *t* and in turn undergoes larger representational decay than the 2nd item, presumably leading to the recency effect. On the other hand, $\sigma_1$ and $\sigma_2$ signify the abstract structure that organizes WM items by assigning different representational precision to items at different positions of a sequence ($\sigma_1$ for the 1st item, $\sigma_2$ for the 2nd item; lower value indicates higher representation precision). Thus, both

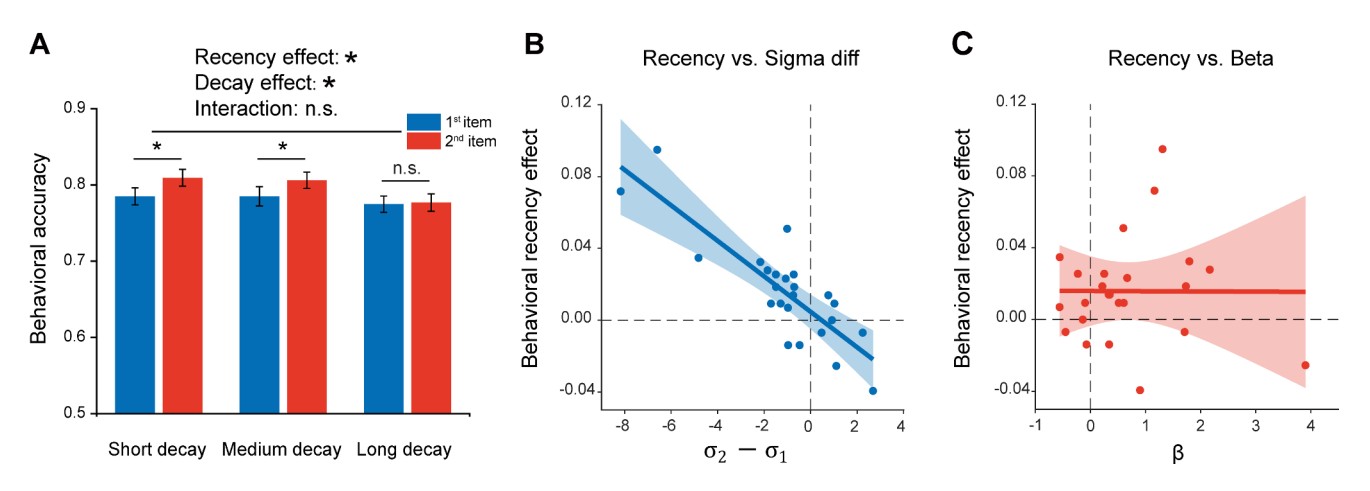

**Figure 5.** Experiment 3 and model fitting. Experiment 3 had the same paradigm as Experiment 1, but with three levels of 2nd orientation-to-PING time intervals so that passive memory decay could be estimated. The model, endowed with sequence structure ($\sigma_1$,$\sigma_2$) and passive memory decay ($\beta$), was used to fit the behavioral data. (**A**) Behavioral results (N = 24). Grand averaged (mean ± SEM) behavioral accuracy for the 1st (blue) and 2nd (red) item at different 2nd orientation-to-PING time intervals. (*: Two-way repeated ANOVA (Recency × Decay), p < 0.05). (**B**) Sequence structure ($\sigma_2 - \sigma_1$) vs. recency effect (partial correlation, r = -0.84, p < 0.001). (**C**) Passive memory decay ($\beta$) vs. recency effect (partial correlation, r = 0.028, p = 0.90). The results support that the recency effect mainly derives from the sequence structure rather than passive memory decay.

The online version of this article includes the following source data for figure 5:

**Source data 1.** Source data for *Figure 5*.

the passive decay and sequence structure would presumably contribute to the recency effect, characterized by $\beta$ and $\sigma_2 - \sigma_1$, respectively.

The computational model was then fitted to the behavioral data to estimate the parameters ($\beta$, $\sigma_1$,$\sigma_2$) in each participant. A partial correlation analysis that calculates the correlation coefficient between two variables by controlling the effect of others, was performed between the estimated parameters ($\beta$, $\sigma_2 - \sigma_1$) and the recency effect across subjects. As shown in *Figure 5*, the sequence structure ($\sigma_2 - \sigma_1$) was significantly correlated with recency effect (*Figure 5B*; r = -0.84, p < 0.001), but not for the decay rate ($\beta$) (*Figure 5C*; r = 0.028, p = 0.90), suggesting that the recency effect is mainly characterized by the sequence structure effect. To further quantitatively test whether the $\beta$ and $\sigma$ differ for two items, we also built two alternative models with one assuming that two items have different $\beta$ and same $\sigma$, and the other assuming that two items have different $\beta$ and different $\sigma$. The group-level Bayesian model selection revealed that the protected exceedance probability for the first model was higher (0.65) than the other two (both 0.17). Therefore, consistent with the MEG findings, the results support a central function of sequence structure, which represents the ordinal information of to-be-memorized items, in mediating the recency effect.

## Discussion

In two MEG experiments, we examined how the sequence structure imposed on a list of WM items shapes their neural representations in the human brain. Items located in different positions of a list are stored in distinct latent states of the WM neural network, being reactivated at different latencies. The reactivation pattern correlates with the recency effect in recognition behavior. In contrast, memorizing the same list of items without sequence structure requirements does not elicit the dissociated reactivations and displays weakened recency effect. Moreover, neural representations of WM items could neither be generalized from the encoding to reactivations nor across items during retention, further advocating the reorganization of items in WM network. Finally, a computational model on the behavioral data supports that the recency effect is mainly derived from abstract sequence structures rather than passive memory decay. Taken together, sequence information, as a form of

abstract structure context, essentially modulates memory performance by reorganizing items into different latent states of the WM neural network.

It has long been posited that WM relies on sustained neuronal firing or activities (*Curtis and D'Esposito, 2003*; *Goldman-Rakic, 1995*; *Vogel and Machizawa, 2004*). Interestingly, recent neural recordings and computational modeling advocate that memory could also be maintained in synapse weights without necessarily relying on sustained activities (i.e. hidden state), through short-term neural plasticity (STP) principles (*Miller et al., 2018*; *Mongillo et al., 2008*; *Rose et al., 2016*; *Sprague et al., 2016*; *Stokes, 2015*; *Trübutschek et al., 2017*; *Wolff et al., 2017*). Recent studies, by developing an interesting impulse-response approach, show that the neural representation of task-relevant features could be successfully reactivated from an activity-silent network (*Wolff et al., 2017*; *Wolff et al., 2020*). Meanwhile, even during the presumably 'activity-silent' period, memory information is still represented in the alpha-band activities (*Bocincova and Johnson, 2019*; *Fahrenfort et al., 2017*; *Sutterer et al., 2019*). Our control analysis based on alpha-band activities during retention supports the view and reveals sustained reactivation profiles (prior to and after the PING stimulus) for both 1st and 2nd orientations (*Figure 3—figure supplement 3*). The active memory representation carried in alpha-band response tends to occur in parallel to the activity-silent storage, that is not disrupted or modified by the PING stimulus. Therefore, two types of mechanism – active, sustained representation and STP-based 'activity-silent' storage –operate together to mediate the working memory process, and future studies could examine their possible distinct functions.

Top-down attention modulates latent states that WM items reside in. For instance, the immediately task-relevant item is in a more excitable state and tends to be first reactivated, compared to ones that are potentially task-relevant in the future (e.g. *Lewis-Peacock et al., 2012*; *Rose et al., 2016*). Notably, WM items in the present study are equally task-relevant and have the same probability to be tested during recalling, and the results thus could not be explained in terms of attentional modulation. In addition to attentional modulation, low-level features could also modulate the excitability of neural populations. A theoretical model posits that multiple items, located in different excitable states according to their bottom-up saliency levels, would be reactivated at different phases within an alpha-band cycle (*Jensen et al., 2012*; *Jensen et al., 2014*). Here, given the serial presentation of items, the lately presented item would presumably have less memory decay and in turn higher saliency level, hence residing in a more excitable state. However, two aspects excluded the explanation. First, Experiment 2 used the same sequence of items as Experiment 1, yet did not reveal the backward profiles. Second, the computational modeling demonstrates that the passive memory decay could not reliably capture the recency effect; rather, it is the sequence structure that plays a central function in modulating recency effect.

Structure information has long been viewed to influence perception and memory, for example global precedence effect (*Chen, 1982*; *Liu et al., 2017*; *Navon, 1977*), reverse hierarchy theory (*Ahissar and Hochstein, 2004*). Recently, it has been suggested that structural and content can be encoded in an independent manner to facilitate memory generalization (*Behrens et al., 2018*; *Bengio et al., 2013*; *Higgins et al., 2017*). Memory performance also tends to be influenced by the contexts the items are embedded in *DuBrow and Davachi, 2013*; *Fischer et al., 2020*; *Gershman et al., 2013*; *Harris, 1952*; *Mathias et al., 2020*. In other words, structure that characterizes the relationship among WM items spontaneously reshapes their neural representations in WM. Our results show that memory representation is reorganized by task contexts, that is ordinal position in Experiment 1 vs. big and small labels in Experiment 2. Meanwhile, sequence structure seems to be a special one and reorganizes WM items into different latent states, which is not the case for other task-relevant dimension, for example big/small labels in Experiment 2. Thus, our results provide new converging behavioral, neural, and modeling evidence advocating the prominent influence of sequence structure on WM behavior and the underlying neural implementations – reorganization of items into different latent states.

The backward reactivation suggests that items located in the late ordinal position are stored in a more excitable state, compared to those in the early positions. The results are well consistent with our previous findings, whereby a temporal response function (TRF) approach was employed to tag the item-specific reactivations (*Huang et al., 2018*),yet indeed differ in important ways. First, the TRF latency refers to relative timing, whereas the present design assessed neural representations in absolute time after the PING, thus providing new evidence for the backward reactivation. Second, the modeling results and Experiment 2 exclude the passive memory decay accounts, an unresolved

issue in previous study. In fact, reversed replay has been observed in many circumstances, for example during break after spatial experience (*Foster and Wilson, 2006*), performing a reasoning task (*Kurth-Nelson et al., 2016*). A recent study shows that during structure learning, a forward replay in spontaneous activities would reverse in direction when paired with reward (*Liu et al., 2019*), implying the involvement of reinforcement learning principles (*Schultz et al., 1997*; *Sutton and Barto, 1998*). Thus, a possible interpretation for the backward reactivation is that the item located in the late position of a list might serve as an anchoring point in memory for other items and is in turn maintained in a more excitable latent state, since the recent item receives less inference and is more reliable for memory retrieval.

Experiment 2 served as a control experiment to test a straightforward alternative explanation (passive memory decay) and has been carefully matched with Experiment 1 in many characteristics, for example same stimuli, WM task, task difficulty. Crucially, in order to make the two experiments comparable in task, subjects recalled orientations according to big/small label. However, the new task would potentially introduce a confounding strategy factor, that is, a direct comparison between the two orientations. Due to the big/small comparisons in Experiment 2, subject might not retain the two orientations precisely as Experiment 1 but just memorize their relative angle or verbal labels. This could possibly account for the less significant reactivations in Experiment2 as well as the comparable reactivations for the two orientations. However, several control analyses weakened the possibility. First, orientation information for both items is indeed precisely represented in terms of the task-relevant dimension (big/small labels) in Experiment 2 (*Figure 4—figure supplement 1*). Second, the behavioral results of Experiment2 are largely accounted for by the angular difference between the to-be-recalled orientation and probe but not by that between the two orientations, confirming that subjects retained two orientations rather than their relative angle in WM. Meanwhile, it remains unknown, at least from the present data, whether comparison task by itself introduced in Experiment 2 would account for the disruption of backward reactivations. Future studies employing a non-comparison task in sequence working memory while controlling other task loads is needed to address the important question.

We built a computational model (*Bays et al., 2009*) that incorporates passive memory decay and sequence structure aiming to understand the recency behavior. The model fitting separates the memory decay influence and confirms that it is the sequence context that mostly accounts for the recency effect. This is also consistent with a previous study revealing that low-level, absolute judgments fail to characterize the high-level, relative judgements (*Ding et al., 2017*). Another explanation of recency effect is the interference account, that is the final item in a list is free from the interference of subsequent items (*Gorgoraptis et al., 2011*), which awaits further studies to investigate. Although the current model only characterized the behavioral performance, the results are highly consistent with the MEG findings, that is decreased recency effect and comparable reactivations when sequence structure context was absent. Taken together, MEG recordings together with the computational model convergingly advocate an essential role of sequence context in WM and its neural implementation, that is reorganizing WM items into different latent states of the neural network.

# Materials and methods

## Participants

Twenty-four subjects (15 males, 21 ± 1.8 years old) participated in Experiment 1, and another 24 subjects (15 males, 21 ± 1.7 years old) participated in Experiment 2. Twenty-five subjects (12 males, 21 ± 2.2 years old) participated in Experiment 3, and one subject was removed due to poor status. All subjects had normal or corrected-to-normal vision, with no history of psychiatric or neurological disorders. All experiments were carried out in accordance with the Declaration of Helsinki. All participants provided written informed consent prior to the start of the experiment, which was approved by the Research Ethics Committee at Peking University (2019-02-05).

## Stimuli and tasks

### Experiment 1

Each trial consisted of three phases – encoding, maintaining, and recalling. During the encoding period, participants were presented with two 0.5 s gratings (6° × 6°) sequentially at the center (0.5 s interval between them), and were instructed to memorize the orientations of the two gratings as well as their order, that is the 1st orientation, the 2nd orientation. For each trial, the orientations of the 1st and 2nd gratings were independently drawn from a uniform distribution over 25°– 175° in steps of 25°, plus a small random angular jitter (± 1° – ± 3°). During the maintaining period, after 1 s, a high luminance disc (30 cd/m$^2$) appeared at the center for 0.1 s, followed by another 1 s interval. During the recalling period, a retrospective cue ('1' or '2' character) was first presented for 1 s to instruct subjects either the 1st or 2nd orientation would be tested. A probe grating (6° × 6°; 20% in contrast, one cycle per degree in spatial frequency, 2 cd/ m$^2$ in mean luminance) was then presented for 0.5 s at the center and participants indicated whether the orientation of the probe was rotated clockwise or anticlockwise relative to that of the cued grating. The angular differences between a memory item and the corresponding memory probe were uniformly distributed across seven angle differences (± 3°, ± 6°, ± 9°, ± 13°, ± 18 °, ± 24°, ± 30°), with 20 trials for each. During each trial, all participants were instructed to keep the number of eye blinks to be minimum. Participants should complete 280 trials in total (determined in a pilot EEG study that used the same number of trials and found successful decoding). The 1st grating was chosen from seven orientation (25° –175° in 25° increments), and each orientation occurred 40 times with random order. The same rule was applied to the 2nd grating. In each trial, the two orientations were drawn independently, but with a constraint that they should at least differ by 25°. It took approximately 40 min (including breaks). Specifically, each subject completed five blocks with each of which containing 56 trials. The grating was chosen from seven orientations and each orientation occurred eight times per block with random order.

### Experiment 2

Experiment 2 had the same stimuli and paradigm as Experiment 1, except that subjects did not need to retain the temporal order of the two orientation features. Specifically, in each trial, subjects were first sequentially presented with two grating stimuli and needed to memorize the two orientations without needing to retain their temporal order as in Experiment 1. During the maintaining period, a high-luminance disc that did not contain any orientation information was presented. During the recalling period, a retro-cue appeared to instruct subjects which item that has either smaller or larger angular values relative to the vertical axis in a clockwise direction ('big' or 'small' in character) would be tested later. Next, a probe grating was presented and participants indicated whether the orientation of the probe was rotated clockwise or anticlockwise relative to that of the cued grating.

### Experiment 3 for modeling (without MEG recording)

This experiment employed the same paradigm as Experiment 1, except that there were three temporal intervals between the offset of the 2nd grating and the PING stimulus (i.e. 1 s, 2.5 s, 4 s). The orientations of the gratings were drawn from a uniform distribution over 20°– 170° in 30° increments, plus a small random angular jitter (± 1° – ± 3°), and the angular differences between a memory item and the corresponding memory probe were uniformly distributed across seven angle differences (± 3°, ± 6°, ± 9°, ± 13°, ± 18°, ± 24°). Similar to Experiment 1, subjects were instructed to memorize the orientations of the two presented gratings as well as their order, that is the 1st orientation, the 2nd orientation. During the maintaining period, a high luminance disc (30 cd/m$^2$) appeared at the center for 0.1 s, followed by another 1 s interval. During the recalling period, a retrospective cue ('1' or '2' character) was first presented for 1 s to instruct subjects either the 1nd or 2nd orientation would be tested. A probe grating (6° × 6°) was then presented for 0.5 s at the center and participants indicated whether the orientation of the probe was rotated clockwise or anticlockwise relative to that of the cued grating. Participants completed 864 trials in total, in three blocks.

## MEG recordings and preprocessing

Participants completed the MEG experiments inside a sound-attenuated, dimly lit, and magnetically shielded room. Stimuli were displayed onto a rear-projection screen (placed at a viewing distance of

75 cm) with a spatial resolution of 800 × 600 pixels and a refresh rate of 60 Hz. Neuromagnetic data were acquired using a 306-sensor MEG system (204 planar gradiometers, 102 magnetometers, Elekta Neuromag system, Helsinki, Finland) at Peking University, Beijing, China. Head movements across sessions should be within 3 mm for data to be involved for further analysis. The spatiotemporal signal space separation (tSSS) was used to remove the external noise (*Taulu and Simola, 2006*). Furthermore, both horizontal and vertical electrooculograms (EOGs) were recorded. MEG data were recorded at 1000 Hz sampling frequency. The MEG data was preprocessed offline using FieldTrip software (*Oostenveld et al., 2011*). Specifically, the data was offline band-pass filtered between 2 and 30 Hz. Independent component analysis was then performed in each subject to remove eye-movement and artifact components, and the remaining components were then back-projected to channel space. All data was then downsampled to 100 Hz. To identify artifacts, the variance (collapsed over channels and time) was first calculated for each trial. Trials with excessive variances were removed. Next, to ensure that each orientation had the same number of trials, we set the minimum number of trials per orientation for decoding analysis to be 37 in each subject, which resulted in at least 259 trials per subject. MEG data was baseline-corrected before further analysis. Specifically, the time range from 500 ms to 0 ms before the presentation of the 1$^{st}$ item in each trial was used as baseline to be subtracted. Since here we focused on the orientation representations in the MEG response, only posterior MEG channels, including parietal sensors (52 planar gradiometers, 26 magnetometers) and occipital sensors (48 planar gradiometers, 24 magnetometers), were used for further analysis.

## Data analysis
### Behavioral performance analysis
In addition to overall behavioral accuracy estimation, to further assess the psychometric function for orientation memory performance, we quantified the response proportion as a function of the angular difference between WM orientation and probe orientation. This function was further fitted in each subject, by $y = 1/(1 + e^{(-\beta(x-\mu))})$, where $\beta$ represents the slope and $\mu$ is the bias parameter. The estimated slope $\beta$ could represent memory precision, with larger value corresponding to better memory performance.

### Time-resolved orientation decoding
To assess the time-resolved orientation information from the MEG signals, we used an inverted encoding model to reconstruct the orientation of the grating stimulus from the neural activities at each time point. This method has been previously used on many features, such as color (*Brouwer and Heeger, 2009*), orientation (*Brouwer and Heeger, 2011*; *Ester et al., 2015*; *Kok et al., 2017*; *Myers et al., 2015*), and spatial location (*Sprague et al., 2014*; *Sprague et al., 2016*; *Sutterer et al., 2019*). One assumption of the model is that the response in each sensor could be approximated as a linear sum of underlying neural populations encoding different values of the feature-of-interest (e.g. orientation) separately, and therefore, by grouping the contributions from many sensors, we could achieve an estimation of the underlying neural population responses.

We began by modeling the response of each MEG sensor as a linear sum of seven information channels. $B_1$ (m sensors × n trials) represents the observed response at each sensor for each trial in the training set. $C_1$ (k channels × n trials) represents the predicted responses of each of the k information channels (i.e. k = 7 here) that are determined by basis functions, for each trial. $W$(m sensors × k channels) represents the weight matrix that characterizes the linear mapping from 'channel space' to 'sensor space'. Taken together, their relationship could be described by a general linear model $B_1 = WC_1$.

Specifically, similar to previous studies (*Brouwer and Heeger, 2011*; *Ester et al., 2015*), the basis functions that would determine $C_1$ are designed to contain seven half-wave rectified sinusoids centered at different orientation values (25°, 50°, 75°, and so on) and raised to the 6th power. The weight matrix $W$(m sensors × k channels) could thus be estimated via using least-squares regression $W = B_1 C_1^T (C_1 C_1^T)^{-1}$.

After establishing $W$ that links sensor space to the underlying information channels responses from the training set, we then used the estimated $W$ to test on independent datasets $B_2$(sensors ×

trials) and calculated the predicted responses of the seven information channels, by $C_2 = \left( W^T W \right)^{-1} W^T B_2$.

The estimated channel responses $C_2$ was then circularly shifted to a common center (0°) in reference to the orientation-of-interest in each trial, which were further averaged across trials. A leave-one-out cross-validation was implemented such that data from all but one experimental block was used as $B_1$ to estimate $W$, while data from the remaining block was used as $B_2$ to estimate $C_2$, to ensure the independence between training set and testing set. The entire analysis was repeated on all combinations, and the resulting information channel responses were then averaged. Note that the procedure was performed at each time point so a time-resolved channel response for each subject was obtained.

To further characterize the orientation decoding performance, the slope of the calculated channel responses at each time was estimated by flipping the reconstruction performance across the center, averaging both sides, and performing linear regression (*Foster et al., 2017*). The slope time courses were further smoothed with a Gaussian kernel (s.d. = 40 ms, *Wolff et al., 2017*).

### Statistical tests for decoding results

As to the reactivation profiles, we first averaged the decoding performance (i.e., slope time course) over trials for each condition. Next, for each condition, one-sample t-test for decoding performance against 0 was performed at each time point. As to the summed reactivation, the slope values of the two conditions were summed and the results were then compared against 0 using one-sample t-test, at each time point. As to the cross-condition reactivation comparison, paired t-test (2nd > 1st, or big > small within Experiment) or independent-sample t test (i.e. difference between Experiments) were performed on the reactivation profiles, at each time point.

A cluster-based permutation test (FieldTrip, cluster-based permutation test, 1000 permutations) (*Maris and Oostenveld, 2007*) was then performed. First, we identified clusters of contiguous significant time points (p < 0.05, two-tailed) from the calculated test statistics (t-value), and cluster-level statistics was calculated by computing the size of the clusters. Next, a Monte-Carlo randomization procedure was used (randomizing data across conditions for 1000 times) to estimate the significance probabilities for each cluster. For cross-condition comparison, condition labels were randomly shuffled between the two conditions. For single condition, 0 (slope value) with the same sample size was generated and shuffled with the original data. The cluster-level statistics was then calculated from the surrogate data to estimate the significance probabilities for each original cluster. All statistical tests were two-sided unless stated otherwise.

### Time-resolved orientation decoding based on big and small labels

We used all the trial types to train the model and then tested big and small labels separately (i.e. fixed encoding model). Specifically, for the 1st orientation which was either labeled as 'big' or 'small' in each trial, we conducted the training on all trials and tested the big- and small-labeled trials, separately. The same analysis was performed on the 2nd orientation. Finally, the big/small orientation decoding results were combined. This analysis ensures that the training dataset contains both big and small orientations with relatively equal probability without being biased to specific orientation range.

### Time-resolved orientation decoding based on alpha-band power

The time-frequency analysis was conducted using the continuous complex Gaussian wavelet transform (order = 4; for example, FWHM = 1.32 s for 1 Hz wavelet; Wavelet toolbox, MATLAB), with frequencies ranging from 1 to 30 Hz, on each sensor, in each trial and in each subject separately, and the alpha-band (8–12 Hz) power time courses were then extracted from the output of the wavelet transform. The decoding analysis was performed on the original alpha-power. We used the same posterior MEG channels selected in previous analysis to perform the alpha-band decoding analysis.

### Correlations between recency behavior and neural reactivation profiles

To evaluate the behavioral relevance of neural reactivation profiles, we computed the Pearson's correlation between the decoding performance and behavioral recency effect, at each time point, on a

subject-by-subject basis. A multiple comparison correction across time was then performed on the correlation results by using cluster-based permutation test (N = 1000). To further illustrate the subject-by-subject correlations between the decoding strength and recency effect (*Figure 3E*), we averaged the decoding strengths over time-of-interests for the 1st (t1: 0.67–0.72 s, after PING) and 2nd (t2: 0.4–0.43 s, after PING) items, respectively, in each subject. The time-of-interests (i.e. t1, t2) were time points showing significant behavioral relevance as well as decoding strength. Note that the time-of-interests were just used for illustration purposes and the correlations between reactivations and the recency effect were independently tested with multiple-comparison correction over time, as described previously (*Figure 3D*).

We used another criterion to choose time-of-interests based on the decoding strength analysis (*Figure 3A*), that is T1 for the 1st item (0.67–0.76 s, after PING) and T2 for the 2nd item (0.26–0.43 s, after PING). The decoding strengths for the 1st and 2nd items were then averaged over T1 and T2, respectively, in each subject. The 2nd – 1st reactivation strength was then correlated with the recency effect (Pearson's correlation; *Figure 3F*).

## Computational modeling

We built a computational model to account for the observed recency effect and the memory decay effect in Experiment 3. The model consists of two sets of parameters ($\beta$ and $\sigma$), which characterize passive memory decay and the contextual influence of ordinal position, respectively (*Figure 5*). For an item, at a given time t after being encoded into working memory, the standard deviation of its representation noise was assumed to be $\sigma + \beta t$ (in three blocks, t = 3.1, 4.6, 6.1, respectively for item 1; t = 2.1, 3.6, 5.1 for item 2). The parameter $\beta$ represents the memory decay rate and the parameter $\sigma$ is the initial standard deviation of orientation representation noise at t = 0, whose value can be $\sigma_1$ or $\sigma_2$ depending on the temporal order of items.

Because of the circular nature of orientation, for a given orientation, the probability distribution of its representation in working memory was assumed to be a von Mises distribution, which is a circular analogue of the familiar Gaussian distribution.

The general form of von Mises distribution is:

$$p(x) = \frac{e^{K\cos(x-u)}}{2\pi \, \text{besseli}(0, K)} \tag{1}$$

Here, x ranges from 0 to $2\pi$. The parameter $u$ is the mean of the distribution and the parameter K is a distribution shape parameter known as 'concentration'.

In our study, the orientation is a circular variable from 0 to $\pi$. In addition, we assumed that the representation precision did not differ for different orientations used in the current study. To simplify the calculation, all orientations of two items, $s_1$ and $s_2$ were set to 0. Thus, the probability distribution of their representation in working memory, p(x) can be written as:

$$p(x) = \frac{e^{K\cos(2x)}}{\pi \, \text{besseli}(0, K)}; x \in \left(-\frac{\pi}{2}, \frac{\pi}{2}\right) \tag{2}$$

Conversion between the von Mises shape parameter K and the standard deviation of representation noise $\sigma + \beta t$, $K = \text{sd2k}(2(\sigma + \beta t))$ is achieved with sd2k function, which is adopted from Bays and his colleagues (*Bays et al., 2009*).

The probe orientation $s_p$ is the relative orientation to the recalled orientation ($s_1$ or $s_2$), ranging from $-\frac{\pi}{2}$ to $\frac{\pi}{2}$ with positive values indicating clockwise compared with the recalled orientation. In each trial, for a given probe $s_p$ (relative orientation), the probability of the binary choice (r = 1, correct; r = 0, wrong) is given by:

$$p(r|s_p) = \begin{cases} \int_{-|s_p|}^{\frac{\pi}{2} - |s_p|} p(x), r = 1 \\ 1 - \int_{-|s_p|}^{\frac{\pi}{2} - |s_p|} p(x), r = 0 \end{cases} \tag{3}$$

Assuming all trials are independent, the joint probability across all N trials can be written as:

$$p\left(r^1|s_p^1\right)p\left(r^2|s_p^2\right)p\left(r^3|s_p^3\right),...,p\left(r^N|s_p^N\right) = \prod_{i=1}^{N} p\left(r^i|s_p^i\right) \tag{4}$$

We next changed products to sums by taking the logarithm of both sides. The log likelihood of our model is given by:

$$\log\left(p\left(r^1|s_p^1\right)p\left(r^2|s_p^2\right)p\left(r^3|s_p^3\right),...,p\left(r^N|s_p^N\right)\right) = \sum_{i=1}^{N} \log\left(p\left(r^i|s_p^i\right)\right) \tag{5}$$

Together, we built a model of three parameters, $\beta$, $\sigma_1$, $\sigma_2$, to quantify the working memory of each item in a sequence. Here, $\beta$ represents the decay rate of memory presentation precision. Parameters $\sigma_1$ and $\sigma_2$ reflect the contextual influence of ordinal position on the representation precision of items in a sequence.

The above model was then fitted to individual behavioral data. For each subject, parameters were estimated to produce the largest value of *Equation (5)* using the Bayesian Adaptive Direct Search (BADS; *Acerbi and Ma, 2017*) with $\sigma_1, \sigma_2 \in [1,50]; \beta \in [-5,5]$.

The above model assumed that the decay rate $\beta$ was same for both items and the $\sigma$ differed depending on their relative ordinal position. To quantitatively test whether the $\beta$ and $\sigma$ differ for two items, we also built two alternative models, one assuming that two items have different $\beta$ and same $\sigma$, and the other assuming that two items have different $\beta$ and different $\sigma$. For each participant and each model, we calculated AIC. The log model evidence was obtained by multiplying AIC by -0.5. The group-level Bayesian model selection was performed using spm_BMS function in SPM 12 (https://www.fil.ion.ucl.ac.uk/spm/software/spm12/). The protected exceedance probability was calculated to indicate which model was more likely than others to describe the data.

## Acknowledgements

We thank Dr. Qing Yu and Dr. Ce Mo for helpful comments. We also thank the three reviewers for their important comments and suggestions in previous submission. This work was supported by the National Natural Science Foundation of China (31930052 to HL), and Beijing Municipal Science and Technology Commission (Z181100001518002 to HL). Dr. Huihui Zhang was supported by Peking University Boya Postdoctoral Fellowship. We also thank National Center for Protein Sciences at Peking University in Beijing, China, for assistance with MEG experiment.

## Additional information

### Competing interests

Huan Luo: Reviewing editor, *eLife*. The other authors declare that no competing interests exist.

### Funding

| Funder | Grant reference number | Author |
| --- | --- | --- |
| National Natural Science Foundation of China | 31930052 | Huan Luo |
| Beijing Municipal Science and Technology Commission | Z181100001518002 | Huan Luo |
| Peking University | | Huihui Zhang |

The funders had no role in study design, data collection and interpretation, or the decision to submit the work for publication.

### Author contributions

Qiaoli Huang, Conceptualization, Formal analysis, Investigation, Visualization, Methodology, Writing - original draft; Huihui Zhang, Investigation, Visualization, Methodology, Writing - original draft;

Huan Luo, Conceptualization, Resources, Supervision, Funding acquisition, Investigation, Methodology, Writing - original draft, Project administration

## Author ORCIDs
Qiaoli Huang ⓘ https://orcid.org/0000-0003-4592-9270
Huihui Zhang ⓘ https://orcid.org/0000-0002-5420-4063
Huan Luo ⓘ https://orcid.org/0000-0002-8349-9796

## Ethics

Human subjects: All experiments were carried out in accordance with the Declaration of Helsinki. All participants provided written informed consent prior to the start of the experiment, which was approved by the Research Ethics Committee at Peking University (2019-02-05).

## Decision letter and Author response
Decision letter https://doi.org/10.7554/eLife.67589.sa1
Author response https://doi.org/10.7554/eLife.67589.sa2

# Additional files

## Supplementary files
• Transparent reporting form

## Data availability
Source data files are provided here: https://osf.io/9amq6/.

The following dataset was generated:

| Author(s) | Year | Dataset title | Dataset URL | Database and Identifier |
|---|---|---|---|---|
| Huang Q, Zhang H, Luo H | 2021 | Sequence structure organizes items in varied latent states of working memory neural network | https://osf.io/9amq6 | Open Science Framework, 9amq6 |

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
