## [Decision Letter]

**Acceptance summary:**

Investigating the role that activity-silent memory plays in supporting sequence information is timely and highly important. The reported core finding is that sequence context organizes working memory items in distinct latent states which can be reactivated during retention. These are compelling findings providing novel insight into how multi-item information is maintained in working memory and how it depends on task context.

**Decision letter after peer review:**

Thank you for submitting your article "Sequence structure organizes items in varied latent states of working memory neural network" for consideration by *eLife*. Your article has been reviewed by 3 peer reviewers, and the evaluation has been overseen by a Reviewing Editor and Laura Colgin as the Senior Editor. The following individual involved in review of your submission has agreed to reveal their identity: Lluís Fuentemilla (Reviewer #2).

Essential revisions:

1) Address the issue of whether the absence of orientation reactivations in Experiment 2 reflect a "reformatting" of the maintained memory representations based on the task demands that is unrelated to whether or not sequence information is stored?

2) Is it possible that orientation information in the present study actually was maintained in an active state, by a signal (e.g. in the alpha-band activity) that was not analysed? This could be investigated anchored in previous work demonstrating successful orientation decoding based on alpha-band activity.

3) The study would benefit from clarity on additional information on implementation on analysis. See below for specifics.

4) A better handling of the specifics of the comparative task presented in Experiment 2 would increase the relevance to other communities interested in how representations (sensory and in working memory) interact over time.

5) In previous work this group has used a metric called representation fidelity to show rhythmic fluctuations of representation of two attended orientations. Could a similar approach be applied here? It can potentially be done even before the ping to investigate for rhythmic fluctuations in the WM representations

*Reviewer #1:*

Huang et al., investigated the role that latent or activity-silent memory states play in supporting memory of sequence information by implementing an impulse response procedure to examine the reactivation profiles of sequentially presented orientations. Applying an inverted encoding model to patterns of MEG data revealed that the second orientation of a two-item sequence was reactivated from an activity-silent state before the first orientation in the sequence when observers were cued to report remembered orientations based on list position. Interestingly, the magnitude of reactivations observed for both items was related to the size of the recency effect observed across participants. In contrast, no difference in reactivation timing was observed when orientation magnitude rather than list position was used to cue the probed orientation. The manuscript concludes that sequence structure is maintained in working memory via the reorganization of information into distinct latent states and that this reorganization is an active process that occurs only when sequence information needs to be maintained.

Overall, the manuscript poses a timely and well-motivated question, and if the reported conclusions hold up to further scrutiny, they provide important insight into how sequences are maintained in working memory. More broadly these observations would also provide another line of evidence in favor of the emerging view that activity silent working memory is a distinct memory process from both active maintenance and long-term memory. The authors did an excellent job including control experiments and analyses as well as being up front about the limitations of their results and assumptions necessary to support their conclusions. Furthermore, the authors make a compelling case that temporal decay does not account for the observed recency effects.

Despite my enthusiasm about many aspects of the paper, the manuscript has several weaknesses that undermine my confidence in its primary conclusions. Specifically, I worry that the absence of orientation reactivations in Experiment 2 reflect a reformatting of the maintained memory representations based on the task demands that is unrelated to whether or not sequence information is stored. Additionally, I worry that orientation information in the present study was maintained in an active state, via a signal (alpha-band activity) that was not analyzed, and that the observed memory reactivations may reflect the contents of this signal rather than a distinct activity-silent signal. I've outlined these points in more detail below.

1. The primary conclusion of the paper is that maintaining order information results in a restructuring of latent memory representations that does not occur when maintenance of sequence information is unnecessary for completing the task. Support for this conclusion rests on the assumption that observers maintain orientation information throughout the entire delay in both experiments, while only the necessity of retaining sequence information differs between experiments. However, the task in Experiment 2 requires a mental comparison between both orientations in order to correctly identify which item in the sequence was a big or small orientation. This comparison process could be done a number of different ways. One way, that is in line with the main conclusion of the manuscript, is to maintain two overlapping orientations throughout the delay. If this strategy was employed, we would expect to see robust reactivations of both orientations after the ping, but no difference in reactivation latency. However, observers could also complete the task by reformatting each orientation representation into a more easily compared code such as a verbal label or point along the edge of the grating as each item is encoded. If this were the case, we would not expect to observe a reactivation of orientation information following the ping. The absence of a significant orientation reactivation for item 1 and observation of only a marginally significant reactivation for item 2 seems to provide stronger support for such a reformatting account than an order free orientation code in Experiment 2.

2. Recent work has shown that multivariate patterns of alpha-band activity (8 – 12 Hz) track centrally (Bocincova and Johnson, 2019; Fahrenfort, Leeuwen, Foster, Awh, and Olivers, 2017) and laterally (Fukuda, Kang, and Woodman, 2016; see Figure 9A) presented orientations as they are maintained in working memory. Thus, there is reason to assume that orientation information in the present study is actively represented throughout the delay-period via alpha-band activity. The primary conclusion of the manuscript is that latent or activity-silent memory states are essential for maintaining sequence information, thus it is important to establish whether or not orientation memory is supported by an active code throughout the delay. Now, it may be possible that a sustained code could operate in parallel with an activity silent code. However, to support the conclusions drawn in the manuscript it is critical to determine whether both codes exist and to provide evidence the observed reactivation effects could not be explained by disruption or reactivation of an active signal in response to the ping stimulus.

References:

Bocincova, A., and Johnson, J. S. (2019). The time course of encoding and maintenance of task-relevant versus irrelevant object features in working memory. Cortex, 111, 196-209.

Fahrenfort, J. J., Leeuwen, J. van, Foster, J., Awh, E., and Olivers, C. N. L. (2017). Working memory implements distinct maintenance mechanisms depending on task goals. BioRxiv, 162537.

Fukuda, K., Kang, M.-S., and Woodman, G. F. (2016). Distinct neural mechanisms for spatially lateralized and spatially global visual working memory representations. Journal of Neurophysiology, 116(4), 1715-1727.

Gorgoraptis, N., Catalao, R. F. G., Bays, P. M., and Husain, M. (2011). Dynamic Updating of Working Memory Resources for Visual Objects. Journal of Neuroscience, 31(23), 8502-8511.

*Reviewer #2:*

The current study investigated how the temporal structure of an encoded sequence of 2 items can be preserved during a short offline period of maintenance in working memory (WM) in humans. The work includes 3 different experimental sets of data and evidence from behavioral, MEG recordings and computational modelling. The results of the study revealed that WM preserves the temporal order of a just encoded sequence of visual two stimuli in "latent states" of neural activity. Latent states are thought to be an efficient way to store information in the brain but require inducing a transient perturbation to be identified empirically. The authors elicited this perturbation after encoding via the visual presentation of a PING stimulus and studied the whether the temporal pattern of reactivation of encoded items during the subsequent maintenance time period. In the first experiment, they showed that second encoded item reactivated before the first item during maintenance and that this reactivation pattern correlated to participants' accuracy in recalling the correct temporal order of the sequence in a subsequent test. To account for the possibility that the backward temporal reactivation profile could not be explained by varied decay strength to the PING stimulus at encoding, they run another experiment and showed that the backward reactivation structure of the sequence disappeared if participants were asked to maintain stimulus properties rather than in their temporal order dimension. Finally, computational modelling on a third data set supported the conclusion that temporal order maintenance was better estimated by parameters of sequential structure than passive memory decay. These data provide strong support to the conclusion that WM preserves the structure of a just encoded experience by reorganizing the neural activity into discrete and temporally separated latent states.

The conclusions of the study are well supported by the data, but some aspects of the data analysis need to be clarified and some mechanistic explanations may benefit from further investigation.

1) Most of the conclusions of the study rely on evidence derived from implementing a time-resolved multivariate analysis (MVPA) on MEG data. While this approach has been successfully used in previous studies with similar stimulation protocols, the study would benefit in clarity if they provided with additional information about its implementation. One such clarification would be to detail how MEG epochs were treated in the analysis. Because of the sequential structure of the encoding and maintenance it would be important to ensure that MVPA analysis preserved always the MEG signal at a specific trial level. It would be important to state if the MEG data was baseline corrected before implementing MVPA and if so, where this baseline period was set, given that maintenance period was preceded by PING stimulus which was in turn preceded by the presentation of the second item of the sequence. A detailed description of the number of trials included per condition and more details on the artifact rejection process seems necessary.

2) An important mechanism thought to support the temporal organization of sequenced items in WM is neural oscillations (i.e., theta and alpha-bands). In fact, the authors highlighted this possibility in the Discussion section, but they fail to explore this possibility on the data. I think the results of the current study would be strengthened if this mechanistic possibility was explored further. Several possible ways can address this issue in their data, given that MEG signal is well suited to this aim. One simple approach would be to investigate the dominance of spectral change modulations in specific frequency bands during the maintenance of temporal structure in experiment 1 but not during the maintenance of stimulus feature properties in experiment 2. One another possibility would be to assess for the existence of temporally structured memory reactivation during maintenance on the bases of MEG oscillatory activity at theta and/or alpha-band. I think that this type of analysis could increase the impact of the paper and extend the interest of the findings to a broader audience in the neuroscientific field.

3) Though the third control study results are in line with findings from experiment 1, they also reveal weaker effects on the 1 sec experimental condition, which is the one that coincides with in experiment 1. The third study however lost power as the number of participants dropped to N = 17 compared to the N = 24 in the first experiment. Given the relevance of the behavioral effects in experiment 1 for this study, I would recommend the authors to increase the sample size of experiment 3 to at least the same number as in experiment 1. That would help evaluate the consistency of the behavioral findings.

4) More details would help understand how the cluster-based permutation test was implemented in the data. How were permutations implemented? Was a cluster identified at the spatial or at the temporal scale?

5) In lines 486-487: " e.g., global precedence effect (Chen, 1982; L. Liu et al., 2017; Navon, 1977), reverse hierarchy theory (Ahissar and Hochstein, 2004), etc." I would recommend avoid using "etc" in the paper as it assumes the reader can make a correct guess about the other examples.

*Reviewer #3:*

This paper provides an innovative technique to the study of sustained activation and reactivation of representations in working memory. Whilst working memory representations have been investigated using the "ping" stimulation in combination with multivariate analyses of non-invasive physiology, this team is applying the same stimulation protocol in conjunction with an inverse encoding model (IEM), not investigated previously. Decoding performance is taken as evidence for reactivation (and from here on will be described as "reactivation markers" or "Activity").

These methodologies are combined with a set of behavioral experiments in order to tie decoding of working memory content to performance. Specifically, the reactivation patterns are used to account for a recency effect in a sequential working memory task.

In experiment 1 after a sequence of two items, a clear reactivation can be found first for the second stimulus of the sequence and later – for the first stimulus in the sequence. Differences in this degree of reactivation are correlated with the behavioral differences in performance between orientation discrimination for the first and second test grating. In this experiment, the items had to be retained for their orientation and order in the sequence. Later on, a probe stimulus would have to be compared to one of the two (cued by its ordinal position).

In a second experiment a near identical stimulation sequence was presented except now subjects were instructed to maintain both items for their angular distance to vertical. Later they were cued to one of the items based on this quantity – they were cued to perform either on the item that had a larger angular distance or the one that has a smaller angular distance. Thus, in addition to being indexed by angular distance to vertical, this task does include an additional element which the authors do not discuss explicitly. In order to select on a given trial which of the two items had a larger angular distance to vertical – a comparison between the items is required.

In this context, arguing that this protocol does not invoke sequence structure (e.g., "without a sequence structure imposed on them") might be overlooking the fact that not only the sequence isn't quite invoked, but it might be perturbed by the active comparison required.

The main findings, and most interesting element the authors provide in this paper is the linking of specific reactivation patterns and behavioral performance in Experiment 1. The methodologies there are solid and if regarded as a control experiment, Experiment 2 indeed points to the fact that task demands likely bring about a consistent reactivation pattern which in turn are related to the behavioral recency effect. The combination of the "ping" manipulation in order to probe latent states of the system is smart and neatly combined with the cutting edge multivariate analysis methods.

The interpretation of Experiment 2, however, are somewhat limited. The assumption of independence of representation for the different items in the sequence (layout above) ends up limiting the scope of analyses. Acknowledging the possibility that the two stimuli are simultaneously processed due to task demands (comparison) could inform interesting analyses that could better delineate the factors that affect working memory reactivation dynamics.

All in all the authors present an innovative combination of methodologies, and shed light on a putative mechanism for recency effects in visual working memory. The scope of the paper most definitely will be of interest to the community researching visual working memory both in human and in animal model. A better handling of the specifics of the "comparative" task presented in Experiment 2 might have increased the relevance to other communities interested in the way in which representations (sensory and in working memory) interact over time.

I would like to just elaborate on ways that could substantiate some of my claims above:

First – perhaps there are idiosynracies in the competitive interaction between wm representations in experiment 2 that require single trial analyses (behavior as well as physiology).

Second, in previous work this group has used a metric called representation fidelity to show rhythmic fluctuations in the representation of the two attended orientations even when coding was not significantly above chance. This could address my above points as well. It can potentially be done even before the ping to look for rhythmic fluctuations in the representation in WM.

[Editors' note: further revisions were suggested prior to acceptance, as described below.]

Thank you for resubmitting your work entitled "Sequence structure organizes items in varied latent states of working memory neural network" for further consideration by *eLife*. Your revised article has been reviewed by 2 peer reviewers and the evaluation has been overseen by Laura Colgin as the Senior Editor, and a Reviewing Editor.

The manuscript has been improved and the additional analysis in the alpha-band has strengthened the results.

There are some remaining issues that need to be addressed. Reviewer 1 has provided a list of remaining concerns. We would like you to address all these concerns of which the overarching issues are:

1) Improve the Methods section in particular on how the alpha-band analysis and how the permutation tests were formed.

2) Some of the claims must be adjusted accordingly to the outcome of the statistical test following the correction for multiple comparisons.

3) The claims on experiment 2 should be moderated in light of the limitations of the experiment (see points 1 and 2).

*Reviewer #1:*

I appreciate the authors' responsiveness to the first round of reviews, and it's clear they put a lot of hard work into running additional analyses and revising the manuscript.

While I think the manuscript is improved as a result, I remain unconvinced that there is unambiguous support for the conclusion that sequence context, rather than task demands more generally, reorganizes the structure of latent memory states. Additionally, a number of the new analyses included in the revision and some of the previously included analyses lack a comprehensive description in the method section, and these missing details make it impossible to evaluate some of the new results that were included in the revision. I've outlined these comments along with others in more detail below:

1. I think the additional analyses included in the revision convincingly rule out the alternative explanation that I raised in my initial review, which is that observers recoded the format of orientation stimuli in Experiment 2 and that such a recoding drives the difference in results across experiments.

2. However, I'm not convinced that these additional analyses rule out the point raised by Reviewer 3, which is that the comparison process that was necessary to complete the task in Experiment 2 was not necessary in Experiment 1. Therefore, it's possible that engaging in this comparison process (even if it doesn't reformat the representations) eliminates the backwards reactivations observed in Experiment 1 and that the backwards reactivation observed in Experiment 1 would occur any time two stimuli are observed in a sequence as long as they don't need to be directly compared. Thus, I'm not convinced that Experiment 2 supports the strong conclusion drawn in the manuscript that the need to store sequence information itself creates a unique latent state for sequence memory. However, I do agree with the authors that the reported work provides support for the more open-ended conclusion that different latent state representations are recruited based on task demands. I think revising the manuscript to emphasize this more modest claim throughout or providing a clear explanation that this limitation of Experiment 2 remains unresolved in the discussion would be an acceptable solution.

3. No details are provided in the method section of the revised manuscript about the alpha analyses that were included in the revision. How was the filtering done? Was the analysis run on total alpha-power as in most previous decoding work or power relative to some sort of baseline (i.e., percent change or DB2). What was the rationale behind using alpha-band power relative to baseline used to identify significant sensors before conducting the multivariate analysis?

4. How was the training and testing conducted for the newly added big and small label analyses? Were all trial types used to train the model and then big and small label orientations were tested separately? If not, using a common training set and then testing on each analysis set separately might improve the power of the analysis (see Sprague et al., 2018 for more details on the benefits of using a "fixed" encoding model). I might be missing something, but it seems like the orientation bin with the smallest angular distance from vertical would never have been included as a "big orientation" trial and the orientation bin with the largest angular distance from vertical would never be included for the "small orientation" decoding is that right? If so, please describe this in the manuscript, and along with how the IEM procedure was adapted to account for these missing orientations.

5. The description of the cluster-based permutation test lines 633 – 639 is still difficult to follow and each step and decision needs to be laid out more clearly. It might help to cite a paper (or papers) that used the same approach applied here and to then report the thresholds set for the current manuscript. For example, some parts of the description sound like you used the approach applied by Sutterer et al. 2019 to identify clusters of above chance orientation selectivity and calculated a null distribution by first shuffling the orientation bin labels and re-running the IEM, repeating this process 1000 times and recording the largest cluster of above chance selectivity observed in each permutation. However, the current description of the analysis states that condition labels, rather than trial labels were shuffled. This approach would enable the identification of clusters where the selectivity differed between two conditions (e.g., big > small orientations), but it's not clear to me how this approach would allow for the identification of clusters of above chance selectivity within a single condition (i.e., when was orientation selectivity for big items higher than expected by chance). Finally, there should be some description of how the permutation test was adjusted for the addition and subtraction analyses. For instance, to calculate the null distribution for the additive analyses that were performed, permuted slope values for item 1 and item 2 (or the big and small orientations) should have been summed before calculating the permuted t-values, were they?

6. The manuscript and response letter refer to effects that were greater than.05 and/or that did not survive multiple comparisons corrections as significant. I know it's frustrating that some of these slopes look "close to significant" or "would have been significant without multiple comparisons correction", but there is no such thing as close to significant in frequentist statistics, so a cluster corrected p value of >.05 should not be reported as significant. The revised manuscript should be adjusted accordingly and arguments resting on the fact that non-significant time courses "look similar" should be removed. Below are a few instances of this that I noticed, but there may be more throughout the manuscript.

a. line 195 – 196 the reactivation of the first item was not significant following cluster correction. This isn't a huge problem since other analyses confirm the relationship between the reactivation of item one and behavioral performance, but it should be reported correctly.

b. Line 258, Decoding performance was not above chance after correcting for multiple comparisons for either item even after changing the permutation test from the two-tailed test used in Experiment 1 to a one-tailed test. Thus, it isn't informative to draw conclusions about whether the reactivation profile of the two items is similar when there is no evidence that each individual item was reactivated in the first place.

c. Line 266 only one of these windows is significant.

7. Line 369- 373: It would be nice to see more of the reasoning that the authors included in the response statement to explain to readers why the existence of this parallel active code does not pose a problem for the activity silent account.

8. Line- 440 (data not shown): show the behavioral analysis as a supplemental figure or report the result in line if it's used to provide support an argument.

References:

Sprague, T. C., Adam, K. C. S., Foster, J. J., Rahmati, M., Sutterer, D. W., and Vo, V. A. (2018). Inverted Encoding Models Assay Population-Level Stimulus Representations, Not Single-Unit Neural Tuning. Eneuro, 5(3), ENEURO.0098-18.2018.

Sutterer, D. W., Foster, J. J., Adam, K. C. S., Vogel, E. K., and Awh, E. (2019). Item-specific delay activity demonstrates concurrent storage of multiple active neural representations in working memory. PLoS Biology, 17(4), e3000239.

*Reviewer #2:*

The authors did a great job and successfully addressed all my previous concerns. Specifically, I think the addition of the results at the alpha-band clearly strengthens the previous findings and makes the paper more appealing to a broader audience.

---

## [Author Response]

Essential revisions:1) Address the issue of whether the absence of orientation reactivations in Experiment 2 reflect a "reformatting" of the maintained memory representations based on the task demands that is unrelated to whether or not sequence information is stored?

We thank the reviewer for raising the important concern, that is, whether the weak reactivation of orientation information in Experiment 2 was due to a “reformatting” of the memory representations given different task demands (making big/small comparison between two orientations in Experiment 2). Specifically, the reviewer worried that “observers could also complete the task by reformatting each orientation representation into a more easily compared code such as a verbal label or point along the edge of the grating as each item is encoded. If this were the case, we would not expect to observe a reactivation of orientation information following the ping”.

This is a very interesting hypothesis, and we believe that several aspects as well as the new analysis results exclude this interpretation. First, we acknowledge that the reactivation for second orientation by itself in Experiment 2 was not significant, but it is noteworthy that the second item still showed a similar trend as that for item 1, and their summed reactivation displayed significant reactivations after PING (Figure 4I), thus to some extent supporting their similar reactivation profiles.

Second and most importantly, we have performed a new analysis to test the possibility raised by the reviewer. Specifically, instead of decoding orientation information based on ordinal position (first or second) as we originally did, we decoded orientation information based on “bigger/smaller” label in Experiment 2. As shown in Figure 4—figure supplement 1B, the Big- and Small-labeled orientations showed significant reactivations with similar temporal profiles after PING (Figure 4—figure supplement 1B; big orientation: 0.14 – 0.23 s, cluster p = 0.042; small orientation: 0.08 – 0.16 s, p < 0.05, uncorrected), and the summation results further support their comparable reactivation patterns (Figure 4—figure supplement 1C; 0.1 – 0.2 s, cluster p = 0.030). Therefore, the orientation information attached to the two labels (Big and Small) is indeed precisely represented in Experiment 2, rather than being encoded as a verbal label or point along a continuum dimension, as worried by the reviewer.

Moreover, we also performed the big/small-labelled decoding analysis during the encoding period. As shown in Figure 4—figure supplement 1A, the two orientations showed overlapping profiles locked to the presented stimuli, which makes sense since the big and small-labeled orientations occurred equally at the two positions. As a control, we also performed the same analysis for Experiment 1. As shown in Figure 4—figure supplement 1, successful decoding was observed during the encoding period (Figure 4—figure supplement 1D), but not during retention (Figure 4—figure supplement 1E, F). Therefore, task demands modulate how the memory system reorganizes orientation representations in the brain, i.e., big/small-labeled orientation decoding in Exp2 (task-relevant) but not in Exp1 (task-irrelevant).

Taken together, the weak orientation reactivations observed in Experiment 2 in the original results are not likely due to a reformatting of memory representation in verbal labels or point along a continuum dimension. Rather, orientation information is still represented precisely but reorganized according to big/small labels, which could be regarded as a type of task-dependent ‘reformatting’.

The new results have been added in Figure 4—figure supplement 1 and Discussion (Page 14-16).

2) Is it possible that orientation information in the present study actually was maintained in an active state, by a signal (e.g. in the alpha-band activity) that was not analysed? This could be investigated anchored in previous work demonstrating successful orientation decoding based on alpha-band activity.

We thank the reviewer for the insightful suggestion. We have performed new analysis on the alpha-band (8-12 Hz) activities throughout the maintaining period to test whether there are parallel active memory representations during retention.

First, we found significant alpha-band activation clusters (compared to baseline) in posterior sensors, during the maintaining period (Author response image 1). We then performed the decoding analysis on the alpha-band power of parietal and occipital sensors (150 sensors in total) throughout the maintaining period. As shown below, both the first and second orientations could be successfully decoded throughout retention in Experiment 1 (Figure 3—figure supplement 3A; see Figure 3—figure supplement 3B for their summed results), thus consistent with previous findings (Bocincova and Johnson, 2019; Fahrenfort, et al., 2017; Fukuda, et al., 2016). Furthermore, the decoding performances showed a relatively sustained pattern (before and after PING stimulus), suggesting that the ‘active memory representation’ carried in alpha-band occurs in parallel to the activity-silent storage and is not disrupted or modified by the PING stimulus.

Furthermore, we did the same alpha-band decoding analysis in Experiment 2, based on big/small labels. Similarly, as shown in Figure 3—figure supplement 3C, D, the orientations could also be successfully decoded in the alpha-band signals and displayed a sustained profile.

We are very grateful for the reviewer’s suggestion which has essentially expanded our findings and advances our understanding about the neural mechanism for sequence working memory. We have added the new results in Figure 3—figure supplement 3 as well as in Discussion (Page 13).

3) The study would benefit from clarity on additional information on implementation on analysis. See below for specifics

We are sorry for the unclear specifications. We have added more details (as below) in Methods now (Page 19-23).

1. The MEG data was first baseline-corrected before decoding analysis. Specifically, the time range from -500 ms to 0 ms relative to the onset of first item in each trial was used as baseline to be subtracted.

2. Each participant in Experiment 1 and Experiment 2 should complete 280 trials in total. The first and the second grating were chosen from 7 orientation (25° –175° in 25° step), and each orientation occurred 40 times with random order. In each trial, the two orientations were drawn independently, but with a constraint that they should at least differ by 25°. The angular differences between a memory item and the corresponding probe were uniformly distributed across seven angle differences (± 3°, ± 6°, ± 9°, ± 13°, ± 18 °, ± 24°, ± 30°). Each angle difference had 20 trials.

3. To identify artifacts, the variance (collapsed over channels and time) was first calculated for each trial. Trials with excessive variances were removed. Next, to ensure that each orientation had the same number of trials, we set the minimum number of trials per orientation for decoding analysis to be 37 in each subject, which resulted in at least 259 trials per subject.

4. About the cluster-based permutation test, we first averaged the decoding performance over all trials belonging to each of the experimental conditions. The test statistics was calculated (t-test) at each time point based on which the significant clusters were found. Cluster-level statistics was calculated by computing the size of cluster. Next, a Monte-Carlo randomization procedure was used (randomizing data across conditions for 1000 times) to estimate the significance probabilities for each cluster.

4) A better handling of the specifics of the comparative task presented in Experiment 2 would increase the relevance to other communities interested in how representations (sensory and in working memory) interact over time.

Thanks for the insightful suggestion. Below we discussed the specifics of the comparative task in Experiment 2.

As acknowledged by the reviewers, Experiment 2 served as a control experiment to test a straightforward alternative explanation, that is, whether the observation is solely due to passive memory decay that arises from sequential presentation of the two orientations. We have carefully matched Experiment 1 and Experiment 2 in many characteristics, e.g., same stimuli, WM task, task difficulty (both around 75%). Crucially, in order to make the two experiments comparable in task demands as close as possible, subjects were instructed to recall orientation information according to big/small label instead of first and second label.

The reviewers raised an interesting concern about the additional comparison task embedded in Experiment 2, i.e., subjects might make a direct comparison between the two orientations rather than memorizing two orientations to achieve the task, which would potentially account for the observations (comparable reactivation profiles).

To address the concern, we fitted a generalized linear mixed-effects model to behavioral performance in Experiment 2, with the angular difference between first and second memory items and that between the to-be-retrieved memory item and probe item as independent variables. We found that only the angular difference between to-be-retrieved memory item and probe accounts for the behavioral performance (β = 0.0013, t = 3.45, p < 0.001), whereas the orientation between memory items could not (β < 0.0001, t = 0.50, p = 0.62). The results suggest that the memory performance in Experiment 2 was not simply mediated by the direct comparison between orientations. Instead, subjects retained two orientations in memory and retrieved the corresponding one and compared it to the probe. The possible reason that subjects did not use a comparison strategy might be the simple task employed here, i.e., relative large angular difference between the first and first orientations. Future studies using more difficult task could address the possibility and potentially reveal different neural reactivation patterns.

Finally, as mentioned in response to the first point, we have performed a new decoding analysis based on big/small labels and revealed significant reactivation for both first and second orientations in Experiment 2 but not in Experiment 1. This supports that task demands indeed modulate how the memory system reorganizes information representations in the brain, a type of task-dependent ‘reformatting’ of sensory information in working memory.

We have added more discussions about the comparison concern in Experiment 2 in Discussion (Page 14-16).

5) In previous work this group has used a metric called representation fidelity to show rhythmic fluctuations of representation of two attended orientations. Could a similar approach be applied here? It can potentially be done even before the ping to investigate for rhythmic fluctuations in the WM representations

We thank the reviewer for the suggestion and have now calculated the representation fidelity as used in our previous study (Mo et al., 2019). As shown in Author response image 2, the fidelity results were actually very similar to the slope index results, during both the encoding and maintaining periods.

**Author response image 2. respfig2:** 

According to the reviewer’s suggestion, we further examined possible rhythmic fluctuations in WM representations during retention. Since the raw results did not show rhythmic profiles (Author response image 2), we calculated the second-to-first cross-correlation coefficient to examine the temporal relationship between the first and second orientation reactivations. Note that rhythmic profiles would predict correlations at periodic time lags. We first performed the cross-correlation analysis for temporal periods before PING (from the disappearance of the second item to the onset of the PING). As in Author response image 3, the time lag between the first and second item was around 0 for both Experiment 1 and 2, suggesting that the two items showed similar temporal profiles. Next, we did the cross-correlation analysis for periods after PING. Interestingly, Experiment 1 showed cross-correlation around 300 ms (Author response image 3) while Experiment 2 showed correlation around 0 (Author response image 3), consistent with the observed sequential reactivations.

Taken together, the results do not reveal rhythmic fluctuations between WM items during retention, suggesting that WM system, especially in terms of the activity-silent state, does not entail rhythmic competition between multiple items, which indeed occurs when multiple features are physically presented and rival for attention (Mo et al., 2019).

**Author response image 3. respfig3:** 

Reviewer #1:[…] Despite my enthusiasm about many aspects of the paper, the manuscript has several weaknesses that undermine my confidence in its primary conclusions. Specifically, I worry that the absence of orientation reactivations in Experiment 2 reflect a reformatting of the maintained memory representations based on the task demands that is unrelated to whether or not sequence information is stored. Additionally, I worry that orientation information in the present study was maintained in an active state, via a signal (alpha-band activity) that was not analyzed, and that the observed memory reactivations may reflect the contents of this signal rather than a distinct activity-silent signal. I've outlined these points in more detail below.1. The primary conclusion of the paper is that maintaining order information results in a restructuring of latent memory representations that does not occur when maintenance of sequence information is unnecessary for completing the task. Support for this conclusion rests on the assumption that observers maintain orientation information throughout the entire delay in both experiments, while only the necessity of retaining sequence information differs between experiments. However, the task in Experiment 2 requires a mental comparison between both orientations in order to correctly identify which item in the sequence was a big or small orientation. This comparison process could be done a number of different ways. One way, that is in line with the main conclusion of the manuscript, is to maintain two overlapping orientations throughout the delay. If this strategy was employed, we would expect to see robust reactivations of both orientations after the ping, but no difference in reactivation latency. However, observers could also complete the task by reformatting each orientation representation into a more easily compared code such as a verbal label or point along the edge of the grating as each item is encoded. If this were the case, we would not expect to observe a reactivation of orientation information following the ping. The absence of a significant orientation reactivation for item 1 and observation of only a marginally significant reactivation for item 2 seems to provide stronger support for such a reformatting account than an order free orientation code in Experiment 2.

Please see our response to the first point in Essential Revision.

2. Recent work has shown that multivariate patterns of alpha-band activity (8 – 12 Hz) track centrally (Bocincova and Johnson, 2019; Fahrenfort, Leeuwen, Foster, Awh, and Olivers, 2017) and laterally (Fukuda, Kang, and Woodman, 2016; see Figure 9A) presented orientations as they are maintained in working memory. Thus, there is reason to assume that orientation information in the present study is actively represented throughout the delay-period via alpha-band activity. The primary conclusion of the manuscript is that latent or activity-silent memory states are essential for maintaining sequence information, thus it is important to establish whether or not orientation memory is supported by an active code throughout the delay. Now, it may be possible that a sustained code could operate in parallel with an activity silent code. However, to support the conclusions drawn in the manuscript it is critical to determine whether both codes exist and to provide evidence the observed reactivation effects could not be explained by disruption or reactivation of an active signal in response to the ping stimulus.References:Bocincova, A., and Johnson, J. S. (2019). The time course of encoding and maintenance of task-relevant versus irrelevant object features in working memory. Cortex, 111, 196-209.Fahrenfort, J. J., Leeuwen, J. van, Foster, J., Awh, E., and Olivers, C. N. L. (2017). Working memory implements distinct maintenance mechanisms depending on task goals. BioRxiv, 162537.Fukuda, K., Kang, M.-S., and Woodman, G. F. (2016). Distinct neural mechanisms for spatially lateralized and spatially global visual working memory representations. Journal of Neurophysiology, 116(4), 1715-1727.Gorgoraptis, N., Catalao, R. F. G., Bays, P. M., and Husain, M. (2011). Dynamic Updating of Working Memory Resources for Visual Objects. Journal of Neuroscience, 31(23), 8502-8511.

Please see our response to the second point in Essential Revision.

Reviewer #2:[…] The conclusions of the study are well supported by the data, but some aspects of the data analysis need to be clarified and some mechanistic explanations may benefit from further investigation.1) Most of the conclusions of the study rely on evidence derived from implementing a time-resolved multivariate analysis (MVPA) on MEG data. While this approach has been successfully used in previous studies with similar stimulation protocols, the study would benefit in clarity if they provided with additional information about its implementation. One such clarification would be to detail how MEG epochs were treated in the analysis. Because of the sequential structure of the encoding and maintenance it would be important to ensure that MVPA analysis preserved always the MEG signal at a specific trial level. It would be important to state if the MEG data was baseline corrected before implementing MVPA and if so, where this baseline period was set, given that maintenance period was preceded by PING stimulus which was in turn preceded by the presentation of the second item of the sequence. A detailed description of the number of trials included per condition and more details on the artifact rejection process seems necessary.

Sorry for previous unclear specifications, and please see our response to the 3^nd^ point in Essential Revision.

2) An important mechanism thought to support the temporal organization of sequenced items in WM is neural oscillations (i.e., theta and alpha-bands). In fact, the authors highlighted this possibility in the Discussion section, but they fail to explore this possibility on the data. I think the results of the current study would be strengthened if this mechanistic possibility was explored further. Several possible ways can address this issue in their data, given that MEG signal is well suited to this aim. One simple approach would be to investigate the dominance of spectral change modulations in specific frequency bands during the maintenance of temporal structure in experiment 1 but not during the maintenance of stimulus feature properties in experiment 2. One another possibility would be to assess for the existence of temporally structured memory reactivation during maintenance on the bases of MEG oscillatory activity at theta and/or alpha-band. I think that this type of analysis could increase the impact of the paper and extend the interest of the findings to a broader audience in the neuroscientific field.

Thank you for the insightful suggestion and we have performed new analysis.

Please see our response to the second point in Essential Revision.

3) Though the third control study results are in line with findings from experiment 1, they also reveal weaker effects on the 1 sec experimental condition, which is the one that coincides with in experiment 1. The third study however lost power as the number of participants dropped to N = 17 compared to the N = 24 in the first experiment. Given the relevance of the behavioral effects in experiment 1 for this study, I would recommend the authors to increase the sample size of experiment 3 to at least the same number as in experiment 1. That would help evaluate the consistency of the behavioral findings.

We have increased the sample size of Experiment 3 to 24, which showed the consistent results as before. The results and Figure 5 have been updated (Page 10-12, 18).

4) More details would help understand how the cluster-based permutation test was implemented in the data. How were permutations implemented? Was a cluster identified at the spatial or at the temporal scale?

Sorry for the unclear description. We first averaged the decoding performance over all trials belonging to each of the experimental conditions. The test statistics was calculated (t-test) at each time point based on which the significant clusters were found. Cluster-level statistics was calculated by computing the size of cluster. Next, a Monte-Carlo randomization procedure was used (randomizing data across conditions for 1000 times) to estimate the significance probabilities for each cluster. Thus, a cluster was identified at temporal scale.

Details have been added in Methods (Page 22-23).

5) In lines 486-487: " e.g., global precedence effect (Chen, 1982; L. Liu et al., 2017; Navon, 1977), reverse hierarchy theory (Ahissar and Hochstein, 2004), etc." I would recommend avoid using "etc" in the paper as it assumes the reader can make a correct guess about the other examples.

Corrected.

Reviewer #3:[…] The interpretation of Experiment 2, however, are somewhat limited. The assumption of independence of representation for the different items in the sequence (layout above) ends up limiting the scope of analyses. Acknowledging the possibility that the two stimuli are simultaneously processed due to task demands (comparison) could inform interesting analyses that could better delineate the factors that affect working memory reactivation dynamics.

We thank the reviewer for the insightful suggestion. We have performed a new decoding analysis based on big/small labels (task-relevant) and revealed significant reactivation for both the first and second orientations in Experiment 2 but not in Experiment 1. The results supports the reviewer’s view, suggesting that task demands indeed modulate how the memory system reorganizes information representations in the brain, a type of task-dependent ‘reformatting’ of sensory information in working memory.

Please also see more details in our response to the first point in Essential Revision.

All in all the authors present an innovative combination of methodologies, and shed light on a putative mechanism for recency effects in visual working memory. The scope of the paper most definitely will be of interest to the community researching visual working memory both in human and in animal model. A better handling of the specifics of the "comparative" task presented in Experiment 2 might have increased the relevance to other communities interested in the way in which representations (sensory and in working memory) interact over time.

Please see our response to the 4^th^ point in Essential Revision.

Comments for the authors:I would like to just elaborate on ways that could substantiate some of my claims above:First – perhaps there are idiosynracies in the competitive interaction between wm representations in experiment 2 that require single trial analyses (behavior as well as physiology).

Thank you for raising the concern. Following this suggestion, we have calculated the first-to-second cross-correlation coefficient in each trial and plotted the distribution of the time lag with the highest correlation coefficient across trials and participants. It is clear that the peak centers around 0, suggesting the two items’ comparable reactivations at single trial level.

**Author response image 4. respfig4:** 

Moreover, since participants recalled one item per trial, we could not examine their possible competition relationship in single trial. We thus examined whether the behavioral performance in Experiment 2 could be accounted for by the angular distance between memory items, by fitting a generalized linear mixed-effects model (Matlab function: fitglme). We found that only the angular difference between the target memory item and probe item accounts for the behavioral performance (β = 0.0013, t = 3.45, p < 0.001), while angular difference between two memory items could not (β < 0.0001, t = 0.50, p = 0.62). This result further supports that memory performance in Experiment 2 does not derive from competition between memory items.Taken together, the observed comparable reactivations for the two orientations is not likely due to the average of a competition profile in single trial.

Second, in previous work this group has used a metric called representation fidelity to show rhythmic fluctuations in the representation of the two attended orientations even when coding was not significantly above chance. This could address my above points as well. It can potentially be done even before the ping to look for rhythmic fluctuations in the representation in WM.

Please see our response to the fifth point in Essential Revision.

[Editors' note: further revisions were suggested prior to acceptance, as described below.]

Thank you for resubmitting your work entitled "Sequence structure organizes items in varied latent states of working memory neural network" for further consideration by eLife. Your revised article has been reviewed by 2 peer reviewers and the evaluation has been overseen by Laura Colgin as the Senior Editor, and a Reviewing Editor.The manuscript has been improved and the additional analysis in the alpha-band has strengthened the results.There are some remaining issues that need to be addressed. Reviewer 1 has provided a list of remaining concerns. We would like you to address all these concerns of which the overarching issues are:1) Improve the Methods section in particular on how the alpha-band analysis and how the permutation tests were formed.2) Some of the claims must be adjusted accordingly to the outcome of the statistical test following the correction for multiple comparisons.3) The claims on experiment 2 should be moderated in light of the limitations of the experiment (see points 1 and 2).Reviewer #1:I appreciate the authors' responsiveness to the first round of reviews, and it's clear they put a lot of hard work into running additional analyses and revising the manuscript.While I think the manuscript is improved as a result, I remain unconvinced that there is unambiguous support for the conclusion that sequence context, rather than task demands more generally, reorganizes the structure of latent memory states. Additionally, a number of the new analyses included in the revision and some of the previously included analyses lack a comprehensive description in the method section, and these missing details make it impossible to evaluate some of the new results that were included in the revision. I've outlined these comments along with others in more detail below:1. I think the additional analyses included in the revision convincingly rule out the alternative explanation that I raised in my initial review, which is that observers recoded the format of orientation stimuli in Experiment 2 and that such a recoding drives the difference in results across experiments.

We are glad that the reviewer is convinced that the new results ruled out the recoding interpretation raised in previous reviews.

2. However, I'm not convinced that these additional analyses rule out the point raised by Reviewer 3, which is that the comparison process that was necessary to complete the task in Experiment 2 was not necessary in Experiment 1. Therefore, it's possible that engaging in this comparison process (even if it doesn't reformat the representations) eliminates the backwards reactivations observed in Experiment 1 and that the backwards reactivation observed in Experiment 1 would occur any time two stimuli are observed in a sequence as long as they don't need to be directly compared. Thus, I'm not convinced that Experiment 2 supports the strong conclusion drawn in the manuscript that the need to store sequence information itself creates a unique latent state for sequence memory. However, I do agree with the authors that the reported work provides support for the more open-ended conclusion that different latent state representations are recruited based on task demands. I think revising the manuscript to emphasize this more modest claim throughout or providing a clear explanation that this limitation of Experiment 2 remains unresolved in the discussion would be an acceptable solution.

Thank you for raising the remaining concern. We agree with the reviewer that our current results so far could not completely exclude another interpretation, that is, the backward reactivation observed in Experiment 1 might be due to the lack of direct comparison task. In other words, Experiment 2 introduced additional task factor, i.e., comparison between memorized items, which might disrupt the backward reactivation observed in Experiment 1.

As suggested by the reviewer, we have added sentences explicitly discussing the limitation of Experiment 2 and future directions in Discussion (Page 16).

3. No details are provided in the method section of the revised manuscript about the alpha analyses that were included in the revision. How was the filtering done? Was the analysis run on total alpha-power as in most previous decoding work or power relative to some sort of baseline (i.e., percent change or DB2). What was the rationale behind using alpha-band power relative to baseline used to identify significant sensors before conducting the multivariate analysis?

We apologize for not including the methods about the alpha-band decoding analysis.

The time-frequency analysis was conducted using the continuous complex Gaussian wavelet transform (order = 4; for example, FWHM = 1.32 s for 1 Hz wavelet; Wavelet toolbox, MATLAB), with frequencies ranging from1 to 30 Hz, on each sensor, in each trial and in each subject separately, and the alpha-band (8 – 12 Hz) power time courses were then extracted from the output of the wavelet transform.

Yes, similar to most previous studies, the decoding analysis was performed on the total alpha-power rather than the relative power change.

About channel selection rationale, we are sorry for the confusion in previous version. In fact, we used the same posterior MEG channels selected in previous analysis to perform the alpha-band decoding analysis.

We have added more details and clarifications in Methods (Page 23 and 24). The figure legend has also been corrected (Figure 3-supplement 3).

4. How was the training and testing conducted for the newly added big and small label analyses? Were all trial types used to train the model and then big and small label orientations were tested separately? If not, using a common training set and then testing on each analysis set separately might improve the power of the analysis (see Sprague et al., 2018 for more details on the benefits of using a "fixed" encoding model). I might be missing something, but it seems like the orientation bin with the smallest angular distance from vertical would never have been included as a "big orientation" trial and the orientation bin with the largest angular distance from vertical would never be included for the "small orientation" decoding is that right? If so, please describe this in the manuscript, and along with how the IEM procedure was adapted to account for these missing orientations.

Thank you for raising the important question.

Yes, we used all the trial types to train the model and then tested big and small labels separately (i.e., fixed encoding model). Specifically, for the first orientation which was either labeled as ‘big’ or ‘small’ in each trial, we conducted the training on all trials and tested the big-labeled and small-labeled, separately. The same analysis was performed on the second orientation. Finally, the big/small orientation decoding results were combined.

The reviewer raised a critical concern that the big and small orientations might always belong to different trials. In fact, if we did training and testing on big- and small-labeled orientations separately, that would constitute an issue that might bias the decoding results. However, as stated above, we used all the trial types (e.g., the first orientation consists of big and small orientations with relative equal probability; same for the second orientation) to train the model and tested big and small labels separately and therefore would not be confounded by the possibility.

We thank the reviewer for the questions and have added more clarifications (Page 23).

5. The description of the cluster-based permutation test lines 633 – 639 is still difficult to follow and each step and decision needs to be laid out more clearly. It might help to cite a paper (or papers) that used the same approach applied here and to then report the thresholds set for the current manuscript. For example, some parts of the description sound like you used the approach applied by Sutterer et al. 2019 to identify clusters of above chance orientation selectivity and calculated a null distribution by first shuffling the orientation bin labels and re-running the IEM, repeating this process 1000 times and recording the largest cluster of above chance selectivity observed in each permutation. However, the current description of the analysis states that condition labels, rather than trial labels were shuffled. This approach would enable the identification of clusters where the selectivity differed between two conditions (e.g., big > small orientations), but it's not clear to me how this approach would allow for the identification of clusters of above chance selectivity within a single condition (i.e., when was orientation selectivity for big items higher than expected by chance). Finally, there should be some description of how the permutation test was adjusted for the addition and subtraction analyses. For instance, to calculate the null distribution for the additive analyses that were performed, permuted slope values for item 1 and item 2 (or the big and small orientations) should have been summed before calculating the permuted t-values, were they?

We apologize for the unclear explanations and have added a separate part with details and analysis steps in Methods (Page 22 and 23).

First, about the test statistics. As to the reactivation profiles, we first averaged the decoding performance over trials for each condition. Next, for each condition (first and second, or big and small), one-sample t-test for decoding performance against 0 was performed at each time point. As to the summed reactivation, the slope values of the two conditions were summed and the results were then compared against 0 using one-sample t-test, at each time point. As to the cross-condition reactivation comparison, paired t-test (second > first, or big > small within Experiment) or independent-sample t test (i.e., reactivation difference between Experiments) were performed on the reactivation profiles, at each time point.

Second, about the cluster-based permutation tests. We first identified clusters of contiguous significant time points (p < 0.05, two-tailed) from the calculated test statistics (specified above). Cluster-level statistics was then calculated by computing the size of cluster. Next, a Monte-Carlo randomization procedure was used (randomizing data across conditions for 1000 times) to estimate the significance probabilities for each cluster. For cross-condition comparison, condition labels were randomly shuffled between the two conditions. For single condition, 0 (slope value) with the same sample size was generated and shuffled with the original data. The cluster-level statistics was then calculated from the surrogate data to estimate significance probabilities for each original cluster.

Finally, we performed the same statistical procedure as Sutterer’s study (Plos Bio., 2019). As shown in Author response image 5, this analysis was actually more sensitive than ours (first item: -0.12 – -0.10 s, cluster p = 0.02, 0.05 – 0.39 s and 0.5 – 0.88 s and 1.13 – 1.4 s, cluster p <0.001; second item: 1.06 – 1.5 s and 1.57 – 1.86 s, cluster p < 0.001).

**Author response image 5. respfig5:** 

6. The manuscript and response letter refer to effects that were greater than.05 and/or that did not survive multiple comparisons corrections as significant. I know it's frustrating that some of these slopes look "close to significant" or "would have been significant without multiple comparisons correction", but there is no such thing as close to significant in frequentist statistics, so a cluster corrected p value of >.05 should not be reported as significant. The revised manuscript should be adjusted accordingly and arguments resting on the fact that non-significant time courses "look similar" should be removed. Below are a few instances of this that I noticed, but there may be more throughout the manuscript.

Thank you for the concern. According to the reviewer’s suggestions, we have revised all the statistical significance statement in a strict and consistent way, i.e., only stating cluster-corrected p valued of < 0.05 to be significant.

Moreover, we would like to emphasize that we have used a conservative criteria (two-tailed test with an alpha level of 0.05) while many studies used one-tailed test.

a. line 195 – 196 the reactivation of the first item was not significant following cluster correction. This isn't a huge problem since other analyses confirm the relationship between the reactivation of item one and behavioral performance, but it should be reported correctly.

Revised (Page 7).

b. Line 258, Decoding performance was not above chance after correcting for multiple comparisons for either item even after changing the permutation test from the two-tailed test used in Experiment 1 to a one-tailed test. Thus, it isn't informative to draw conclusions about whether the reactivation profile of the two items is similar when there is no evidence that each individual item was reactivated in the first place.

Revised (Page 9).

c. Line 266 only one of these windows is significant.

Revised (Page 10).

7. Line 369- 373: It would be nice to see more of the reasoning that the authors included in the response statement to explain to readers why the existence of this parallel active code does not pose a problem for the activity silent account.

Added (Page 13).

8. Line- 440 (data not shown): show the behavioral analysis as a supplemental figure or report the result in line if it's used to provide support an argument.

Added (Page 9).

References:Sprague, T. C., Adam, K. C. S., Foster, J. J., Rahmati, M., Sutterer, D. W., and Vo, V. A. (2018). Inverted Encoding Models Assay Population-Level Stimulus Representations, Not Single-Unit Neural Tuning. Eneuro, 5(3), ENEURO.0098-18.2018.Sutterer, D. W., Foster, J. J., Adam, K. C. S., Vogel, E. K., and Awh, E. (2019). Item-specific delay activity demonstrates concurrent storage of multiple active neural representations in working memory. PLoS Biology, 17(4), e3000239.